# Water Clusters in Interaction with Corannulene in a Rare Gas Matrix: Structures, Stability and IR Spectra

Heloïse Leboucher [1], Joëlle Mascetti [2], Christian Aupetit [2], Jennifer A. Noble [3] and Aude Simon [1,*]

1    Laboratoire de Chimie et Physique Quantiques (LCPQ), Fédération FeRMI, UMR5626, Université de Toulouse [UT3] & CNRS, University Paul Sabatier, 118 Route de Narbonne, 31062 Toulouse, France; heloise.leboucher@irsamc.ups-tlse.fr
2    Institut des Sciences Moléculaires (ISM), Université de Bordeaux, UMR5255 CNRS, 351 Cours de la Libération, CEDEX, 33405 Talence, France; joelle.mascetti@u-bordeaux.fr (J.M.); christian.aupetit@u-bordeaux.fr (C.A.)
3    Laboratoire Physique des Interactions Ioniques et Moléculaires (PIIM), UMR7345, CNRS, Aix-Marseille Université, 13013 Marseille, France; jennifer.noble@univ-amu.fr
*    Correspondence: aude.simon@irsamc.ups-tlse.fr

**Abstract:** The interaction of polycyclic aromatic hydrocarbons (PAHs) with water is of paramount importance in atmospheric and astrophysical contexts. We report here a combined theoretical and experimental study of corannulene-water interactions in low temperature matrices and of the matrix's influence on the photoreactivity of corannulene with water. The theoretical study was performed using a mixed density functional based tight binding/force field approach to describe the corannulene-water clusters trapped in an argon matrix, together with Born-Oppenheimer molecular dynamics to determine finite-temperature IR spectra. The results are discussed in the light of experimental matrix isolation FTIR spectroscopic data. We show that in the solid phase, $\pi$ isomers of $(C_{20}H_{10})(H_2O)_n$, with $n$ = 2 or 3, are energetically favored. These $\pi$ complexes are characterized by small shifts in corannulene vibrational modes and large shifts in water bands. These $\pi$ structures, particularly stable in the case of the water trimer where the water cluster is trapped "inside" the corannulene bowl, may account for the difference in photoreactivity of non-planar–compared to planar–PAHs with water. Indeed, planar PAHs such as pyrene and coronene embedded in $H_2O$:Ar matrices form $\sigma$ isomers and react with water to form alcohols and quinones under low energy UV irradiation, whereas no photoreactivity was observed for corannulene under the same experimental conditions.

**Keywords:** PAHs; astrochemistry; matrix isolation FTIR spectroscopy; water clusters; corannulene; DFTB; BOMD

## 1. Introduction

The interaction of polycyclic aromatic hydrocarbons (PAHs) with water is of paramount importance in a large number of fields, such as the petroleum industry [1], atmospheric science [2,3], astrophysics and astrochemistry. Due to its allotropy, a large fraction of interstellar carbon is expected to be included in large molecules or in dust grains with various degrees of organization and aromaticity, including fullerenes, diamondoids, PAHs, clusters of PAHs, or amorphous carbon grains, possibly hydrogenated [4]. AstroPAHs in particular are believed to be present in the interstellar medium (ISM), where they account for up to 20% of the total carbon [5,6]. PAHs have been of significant interest since the proposal in the eighties that they were the carriers of the Aromatic Infrared Bands (AIBs), a set of mid-IR emission bands observed in many regions of the ISM [7,8]. This proposal led to many experimental and theoretical spectroscopic studies in order to identify a specific PAH molecule [6]. However, it is only very recently that specific PAH molecules, the two isomers of cyano-naphthalene, have been successfully detected based on their rotational spectra [9].

In interstellar cold dense molecular clouds, where the temperature ranges from 10 to 50 K, PAHs are expected to be frozen on or in icy mantles of dust grains [10]. These ice

mantles are mainly composed of amorphous solid water (ASW), and allow chemical reactions to occur after heating or irradiation [11]. It has been shown that PAHs embedded in ASW, when irradiated by VUV light, undergo erosion processes and lead to oxidation products such as alcohols and ketones [12] via the formation of cationic intermediates. Interestingly, the formation of oxygenated PAHs is also observed in ASW when using low energy UV-visible irradiation [13–16]. It has also been shown that small complexes of coronene and water isolated in cryogenic argon matrices exhibit photochemical reactions similar to those observed in ASW [17]. We have previously demonstrated that, when embedded in a solid matrix, coronene forms $\sigma$ complexes with water molecules arranged in interaction with the edge of the planar PAH [18]. Such $\sigma$ interaction is expected to favor photochemical reaction through better H/charge transfer to form oxidation products from planar PAHs [17,19]. These results suggest that a relatively easy formation of oxygenated PAH photoproducts could account for shifts observed in the AIBs.

In this work, we present some experimental and theoretical results concerning corannulene $C_{20}H_{10}$ in interaction with water clusters embedded in a rare gas matrix. Corannulene is a fully aromatic non-planar PAH [20] of $C_{5v}$ symmetry, consisting of a five membered carbon cycle surrounded by five six-membered carbon cycles that can be described as a bowl-like molecule with two faces: convex and concave (see Figure 1).

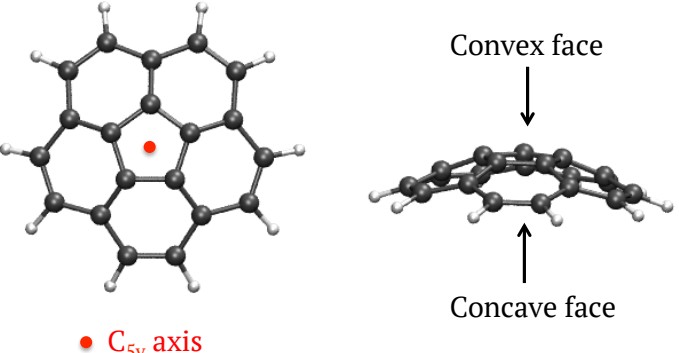

**Figure 1.** Scheme of corannulene $C_{20}H_{10}$, a hydrogenated cap of buckminsterfullerene of $C_{5v}$ symmetry, top and side views.

Corannulene can be regarded as a cap of buckminsterfullerene $C_{60}$—a molecule that was identified in Space more than 10 years ago [21,22]—saturated with H atoms. It can thus reasonably be considered as a possible intermediate in "top-down" or "bottom-up" routes leading to $C_{60}$ formation. The mechanism for the formation of buckminsterfullerene $C_{60}$ is still under debate. Following its discovery, observational studies suggested that fullerenes could be produced by photochemical dehydrogenation and isomerisation of amorphous carbon grains [23–26], but the composition of such grains remains elusive. Among current hypotheses, buckminsterfullerene could be formed following decay and rearrangement of dehydrogenated PAHs [27] under the action of UV photons [28] or by collisions with energetic particles [29]. Such a "top-down" mechanism is expected to prevail in low-density environments such as interstellar clouds [30] and the corannulene molecule can be regarded as a possible intermediate in such a sequence. Alternatively, a "bottom-up" mechanism, consisting of the successive growth from smaller carbon chain building blocks, known as the closed network growth mechanism [31], would prevail in the hot, dense envelopes of evolved stars [30]. The need to characterize the structural diversity of carbon clusters, possibly hydrogenated and with amorphous character, that could play a role in the formation of buckminsterfullerene, motivated the development of dedicated experimental and theoretical approaches in order to obtain their spectral features in the UV-visible [32,33] and IR domains [34–37].

Because of its bowl shape, corannulene could be detected in astrophysical environments by radioastronomy, as put forward by Pilleri et al. [38]. The astrophysical interest of corannulene has also motivated experimental and theoretical spectroscopic studies in the IR, Raman and UV-visible domains [39], both in the solid phase and, in particular, in rare

gas matrices [39], in order to propose spectroscopic characteristics that should be searched for in space. Recently, its formation in the gas phase was achieved by Zhao et al. [40] using a "bottom-up" approach involving the reaction of 7-fluoranthenyl ($C_{16}H_9$) and benzo[ghi]fluoranthen-5-yl ($C_{18}H_9$) radicals with acetylene ($C_2H_2$) in conditions mimicking those of carbon-rich circumstellar envelopes. Interestingly, due to its bowl shape that can invert and form host-guest complexes, corannulene also presents a particularly interesting building block in materials chemistry [41].

Studies on the structures, energetics and spectroscopic signatures of corannulene-water complexes are, to the best of our knowledge, scarce. Theoretical studies of water clusters $(H_2O)_n$ in interaction with $[C_{20}H_{10}]^+$ were performed by Hernández-Rojas et al. [42]. Global optimization using a force field approach was used to determine the geometries of the larger clusters. They showed that, for $n = 1$ and 2, the water cluster preferentially binds to the side of $[C_{20}H_{10}]^+$, whereas larger sized water clusters interact preferentially with the concave face of cationic corannulene. Gas phase neutral corannulene-water complexes have been studied by Pérez et al. [43] using rotational spectroscopy complemented by *ab initio* calculations. They have shown that in $(C_{20}H_{10})(H_2O)$, the water molecule resides inside the bowl-like structure of corannulene, where it rotates freely around the $C_5$ axis and that dispersion interactions represent the main contribution to the binding. We have decided to investigate $(C_{20}H_{10})(H_2O)_n$ complexes isolated in argon matrices at 10 K. Our results are compared to those obtained in the gas phase [43] and to our previous study on coronene:water complexes. We detail our theoretical and experimental approaches in Section 2, report our results in Section 3, and discuss the experimental results in light of the calculations in Section 4.

## 2. Materials and Methods

### 2.1. Computational Methods

In order to determine the structures, energetics and IR spectra of $(C_{20}H_{10})(H_2O)_n$ ($n = 1$–3) embedded in an argon matrix, (hereafter quoted as: Ar) we used the same approach as that previously applied to coronene-water clusters $(C_{24}H_{12})(H_2O)_n$ ($n = 1, 2$) [18]. All calculations were performed with the deMonNano code [44].

The electronic structure of $(C_{20}H_{10})(H_2O)_n$ is described at the self consistent charge density functional based tight binding (SCC-DFTB) level of theory [45] using the Hamiltonian that we modified so as to obtain a good description of PAH-water interactions and used to describe coronene:water aggregates in an Ar matrix [18]. Briefly, an empirical dispersion term was added and the Mulliken charges were replaced by Truhlar Charge Model 3 (CM3) charges [46] in order to improve the description of the polarisation of the bonds. The latter modification revealed important for the description of long range electrostatic interactions [47,48] and for IR spectra [49,50]. This modified SCC-DFTB scheme (denoted hereafter simply as DFTB) was benchmarked against wavefunction and DFT calculations [48,51]. It was shown to provide a good description of the structures and energetics of water clusters and PAHs in interaction with water clusters [48,51,52]. The values of the DFTB shifts induced by the adsorption of PAHs determined for the IR bands of water-ice dangling OH were shown to be in good agreement with experimental data [53]. Focusing more specifically on corannulene:water, when optimising the structure of $(C_{20}H_{10})(H_2O)$ at the DFTB level, we found that the most stable structure was similar to that found by Pérez et al. [43] in the gas phase, i.e., with the water molecule interacting with the concave face of corannulene. We computed a DFTB intermolecular binding energy of $-21 \, kJ \cdot mol^{-1}$ (without zero-point energy) which is in very good agreement with the binding energy of $-22.40 \, kJ \cdot mol^{-1}$ determined using symmetry adapted perturbation theory (SAPT) analysis [43]. Besides, additional comparison between gas phase DFTB and DFT structures, energetics and IR spectra for isolated corannulene:water clusters was made (see data base associated to the present paper), revealing similar trends as for benzene:water and coronene:water clusters [48].

The argon-argon interaction is described at the force field (FF) level using the Aziz potential [54]. The DFTB/FF coupling is incorporated via a scheme close to first order degenerate perturbation theory and developed to describe clusters of hydrocarbons and water

trapped inside a rare gas matrix [55,56]. This DFTB/FF scheme was carefully benchmarked for PAH:H$_2$O:Ar systems, i.e., involving the same interactions as in this work, regarding structures, energetics and IR spectra [18,55,56]. In this work, the Ar matrix is modeled by a large cluster (1139 Ar atoms for the total pure Ar matrix) organised in a face cubic centered (fcc) crystalline structure at the center of which the impurity (($C_{20}H_{10}$)(H$_2$O)$_n$) is inserted after removing a minimal number x of Ar atoms (x is the vacancy size). In the rest of the manuscript, the total system will be referred to as ($C_{20}H_{10}$)(H$_2$O)$_n$:Ar.

To obtain the initial geometries, several symmetry planes were considered for inserting the corannulene molecule into the matrix. After determining the most stable situation for the embedded corannulene, all the possible interaction sites of the water molecule with $C_{20}H_{10}$ were considered. For *n* = 1, the substitution of one Ar atom by one water molecule was considered for all Ar atoms interacting with the concave and convex faces of corannulene as well as in the [111] insertion layer of corannulene when Ar atoms interact with the H atoms of corannulene. Once the most stable geometries for the three situations are determined, the substitution of all possible Ar atoms interacting with the water monomer in the most stable ($C_{20}H_{10}$)(H$_2$O)$_n$:Ar systems by another water molecule are considered. The most stable situations where the dimer interacts with the concave face, convex face, and "in plane" H atoms of corannulene are retained and the interaction of a third water molecule with the existing dimer is considered. Local geometry optimisations are then performed and the relative stabilities of the determined isomers are estimated by comparison of their substitution energy values ($E_{sub}$, defined in ref. [18]) determined as follows: $E_{sub}$ is equal to the energy of the optimised total system ($E$($C_{20}H_{10}$)(H$_2$O)$_n$:Ar)) plus the energy of the vacancy ($E$(Ar$_x$)) minus the energy of the optimised total matrix ($E$(:Ar)) minus the energy of the isolated monomers ($E$($C_{20}H_{10}$) + $nE$(H$_2$O)), i.e.,

$$E_{sub} = E((C_{20}H_{10})(H_2O)_n : Ar) + E(Ar_x) - E(: Ar) - E(C_{20}H_{10}) - nE(H_2O) \quad (1)$$

The IR spectra of the most stable (lowest value of $E_{sub}$) energy structures for all water cluster sizes in all three configurations in which the water cluster interacts with the concave face, convex face and "in plane" H atoms were first computed in the harmonic approximation. However, finite-temperature effects due to the anharmonicity of the PES may be expected even at low temperature. In this case, an *a priori* more suitable approach is to derive IR spectra from molecular dynamics (MD) simulations computing the Fourier transform of the dipole moment $\mu$ autocorrelation function

$$\alpha(\omega) \quad \propto \quad \omega^2 \int_0^{+\infty} dt \langle \mu(0) \cdot \mu(t) \rangle \, e^{i\omega t} \quad (2)$$

Therefore we used the same approach as in the work by Simon et al. [18], consisting in performing Born-Oppenheimer MD (BOMD) simulations in the microcanonical ensemble, computing the electronic structure on-the-fly with the DFTB/FF method. The IR absorption cross sections are determined using Equation (2) with the electrostatic dipole moment computed using the CM3 charges of the corannulene:water clusters, the Ar atoms remaining neutral. The positions of the Ar atoms at the surface of the cluster are frozen to ensure the absence of surface Ar atom motions in the molecular dynamics. The system was first thermalised at the desired temperature (10 K) by means of short simulations (10 ps) in the canonical ensemble using a Nose-Hoover chain of thermostats. These first (NVT) simulations allow the generation of the initial conditions for MD simulations performed in the microcanonical ensemble (NVE). For each corannulene-water cluster, 7 MD simulations of 100 ps in the (NVE) ensemble were run, using a time step of 0.1 fs. The IR spectra were computed for each simulation and averaged. In such conditions (low temperature), convergence is expected on positions but not necessarily on intensities [18], which was checked for the specific systems studied in the present work (see Appendix B, Figure A1). In addition, the quality of the dynamic IR spectra inside the Ar matrix was checked for the water monomer and dimer which, as will be discussed in Section 3.1.2, rotate in the matrix,

increasing the simulation time from 100 ps to 500 ps (see Appendix B, Figure A2). Due to the large number of large systems to be investigated consistently, we used the systematic procedure described hereabove (7 × 100 ps) for all of them.

### 2.2. Experimental Methods

Experiments were performed in a high vacuum experimental setup consisting of a stainless steel chamber with a base pressure of $10^{-7}$ mbar, containing a CsBr substrate cooled to 10 K by a closed-cycle He cryostat (Cryophysics Cryodine). The sample temperature was monitored by a Si diode thermometer positioned on the copper substrate holder in proximity to the CsBr window. Deionised ultrapure water was subjected to multiple freeze-pump-thaw cycles under vacuum to remove dissolved gases. Argon and water were premixed in a dosing line with concentrations of water in argon varying from 1:10 to 1:1 and injected into the chamber at a rate of 1 mL min$^{-1}$ in order to form different mixtures containing $H_2O$ monomers, dimers, trimers, and larger clusters up to hexamers. Corannulene (TCI, 97%, used without purification) was sublimated by heating the powder to 120 °C in an oven within the high vacuum chamber and co-deposited with the argon:water mixture which was introduced as a diffusion jet flow via a gas nozzle inlet located 2 cm from the CsBr substrate. Depositions lasted for between one and two hours. Infrared spectra of corannulene:$H_2O$:Ar samples were recorded in transmission mode using a Bruker 70 V FTIR spectrometer with a DTGS detector from 4000 to 400 cm$^{-1}$ at a 0.5 cm$^{-1}$ resolution, with each spectrum averaged over 200 scans. Reference spectra of corannulene:Ar and $H_2O$:Ar at different concentrations were also recorded for comparison. All spectra were recorded at 10 K. Depositions were irradiated at 10 K with a mercury lamp ($\lambda > 235$ nm, average power 150 mW, total fluence 0.72 J m$^{-2}$), and IR spectra measured after a series of irradiations (from 1 to 120 min).

### 3. Results

#### 3.1. Computational Results: Influence of the Matrix

3.1.1. Structures and Energetics

Corannulene was inserted in different symmetry planes of the Ar matrix represented by a finite size fcc cluster initially of 1139 Ar atoms. After removing the minimum number of Ar atoms we found that the most stable situation corresponded to an insertion of the corannulene molecule in the [111] plane with a vacancy of 7 Ar atoms (see Figure 2, $E_{sub} = -2470$ cm$^{-1}$, also see Appendix A, Table A1 for the results of the other symmetry planes). We obtained the same result as for coronene [18] and other PAHs such as naphthalene [57]. The substitution energy is smaller than that computed for coronene using the same model ($E_{sub} = -4810$ cm$^{-1}$ [18]) which can be understood as the non planearity of corannulene leads to more perturbation of the Ar layers close to corannulene.

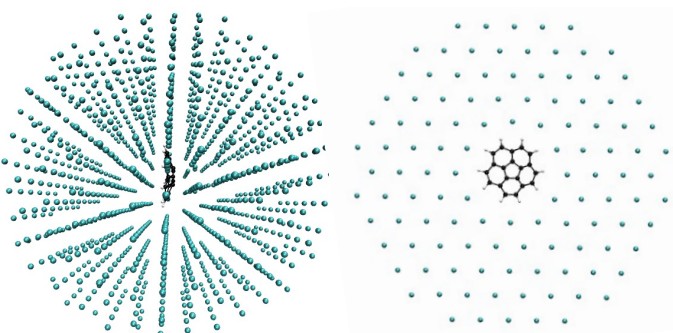

**Figure 2.** Optimised structure of $C_{20}H_{10}$:Ar obtained at the DFTB/FF level of theory. The total system is reported on the left hand side, the selection of the [111] plane where corannulene is inserted is represented on the right hand side.

In Table 1 are reported the substitution energies obtained using Equation (1) for the most stable isomers of $(C_{20}H_{10})(H_2O)_n$:Ar ($n$ = 1–3) of different types i.e., depending

whether the water molecules interact with the concave or convex faces of corannulene or with its H atoms. The geometries of the most stable isomers are reported in Figures 3–5 for $n = 1, 2$ and 3 respectively.

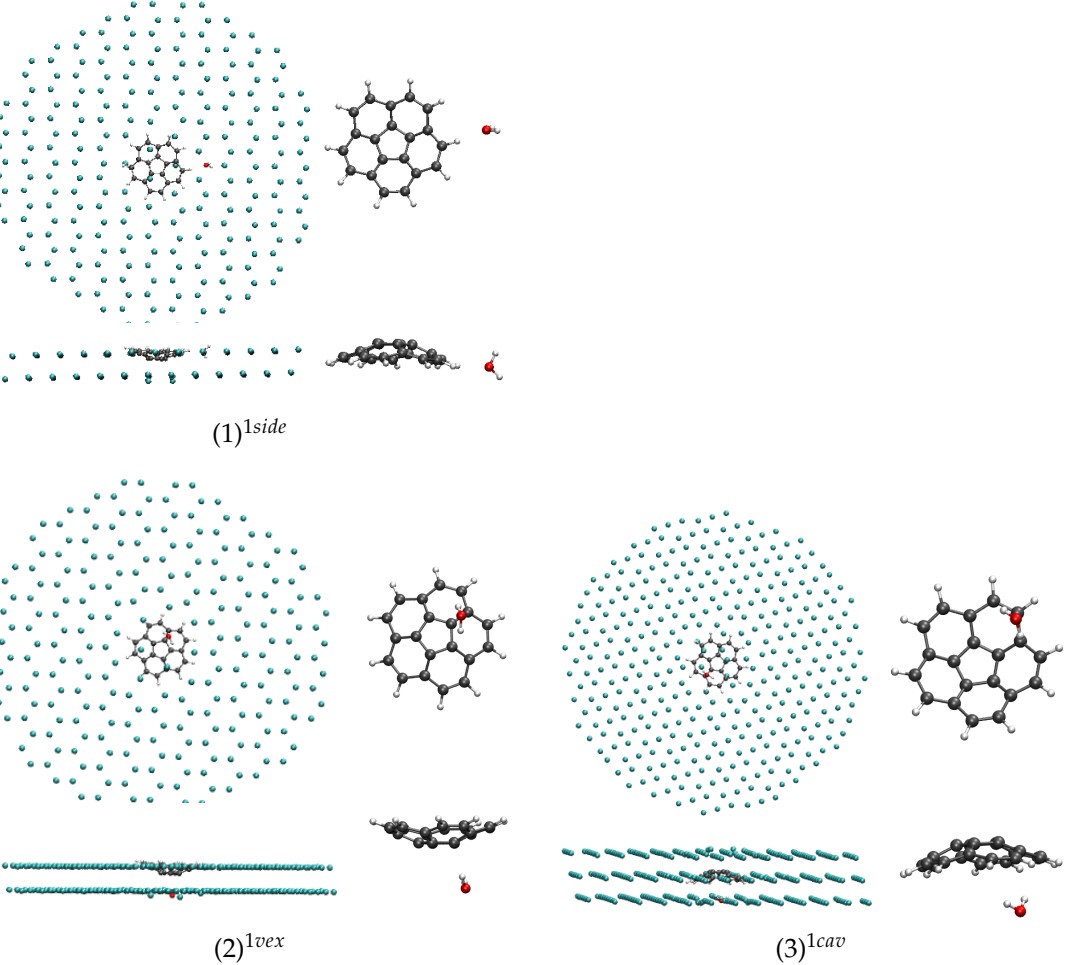

**Figure 3.** Most stable structures for $(C_{20}H_{10})(H_2O)$:Ar obtained at the DFTB/FF level of theory (selection of layers).

In the case of $n = 1$, the three conformers $(1)^{1side}$, $(2)^{1vex}$ and $(3)^{1cav}$ lie in an energy range of 150 cm$^{-1}$ (1.8 kJ· mol$^{-1}$), the most stable one corresponding to the water molecule interacting with an H atom of corannulene in the [111] symmetry plane $((1)^{1side})$. Interestingly, the conformer in which the water molecule interacts with the concave face, known to be the most stable isomer in the gas phase [43], is the highest energy one $((3)^{1cav})$, although the energy difference is small. These subtle differences from the gas phase originate from the constraints of the fcc lattice. For instance, in the $(3)^{1cav}$ isomer, the water molecule substitutes an Ar atom whose position is not aligned with the $C_{5V}$ axis of corannulene, and this prevents the water molecule from having the most stabilizing interaction with the corannulene as in the gas phase. When relaxing the geometry of this isomer after removing the matrix environment, the water molecule migrates towards the center of the "cup" and becomes the lowest energy isomer (see last column of Table 1).

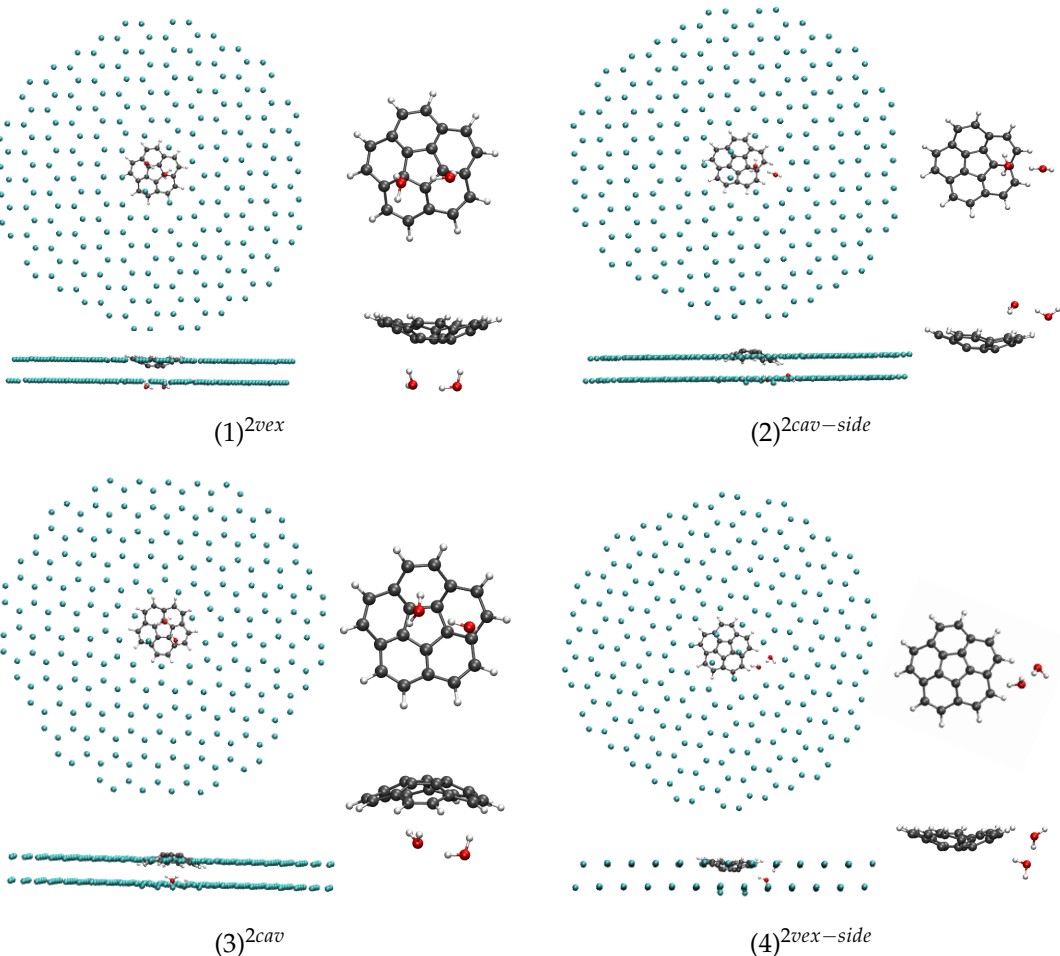

**Figure 4.** Most stable structures for $(C_{20}H_{10})(H_2O)_2$:Ar obtained at the DFTB/FF level of theory (selection of layers).

The perturbation of the [111] Ar layers adjacent to that containing corannulene due to the presence of the impurity can be visualized on the side views of the systems in Figures 3–5, a few Ar atoms being moved out of their original plane. Now comparing with the results obtained for coronene ($C_{24}H_{12}$, planar PAH) inside an Ar matrix using the same computational approach, the isomer similar to the $(1)^{1side}$ isomer of the present work, designated as $\sigma$ isomer in Simon et al. [18], was found more stable than the $\pi$ isomer by $\sim$800 cm$^{-1}$ in the case of coronene. This means that the stabilization of the "side" (or "$\sigma$") isomer is enhanced in the case of a planar PAH with respect to the case of corannulene where it appears in competition with the two "$\pi$" isomers $(2)^{1vex}$ and $(3)^{1cav}$.

In the case of $n$ = 2, the most stable isomer corresponds to the water dimer interacting through two H atoms with the convex face of corannulene ($(1)^{2vex}$, see Figure 4), the two H atoms pointing towards two central C atoms of corannulene. The corresponding isomer in which the water dimer interacts with the concave face of corannulene ($(3)^{2conv}$) is found 210 cm$^{-1}$ (or 2.5 kJ·mol$^{-1}$) higher in energy. The isomer in which the water dimer interacts with both the concave (resp. convex) face and side H of corannulene is found 190 cm$^{-1}$ or 2.3 kJ·mol$^{-1}$ (resp. 240 cm$^{-1}$ or 2.9 kJ·mol$^{-1}$) higher in energy. As in the case of $n$ = 1, when the geometries of $(C_{20}H_{10})(H_2O)_2$ are further relaxed after removal of the Ar matrix, the water dimer tends to move towards the center of the cup when it interacts with the concave face of corannulene and it leads to the stabilisation of such conformers (see last column of Table 1).

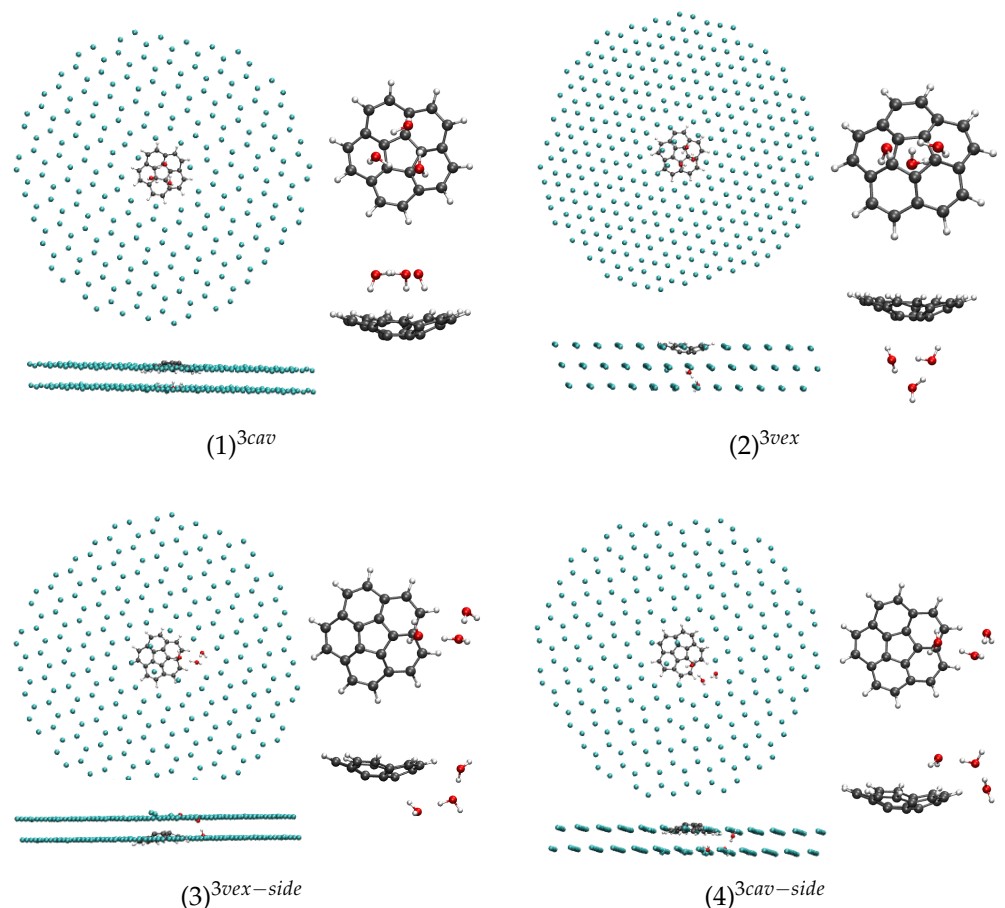

**Figure 5.** Most stable structures for $(C_{20}H_{10})(H_2O)_3$:Ar obtained at the DFTB/FF level of theory (selection of layers).

**Table 1.** Substitution energies of the $(C_{20}H_{10})(H_2O)_n$ complexes ($n \leq 3$) and, for a given stoichiometry, relative energy with respect to the most stable system. In parenthesis are added the relative energy of the corresponding isolated $(C_{20}H_{10})(H_2O)_n$ aggregate (after removing the Ar matrix) after local DFTB geometry optimisation.

| $n_{H_2O}$ | Number of Atoms | Isomer | $E_{sub}$ (cm$^{-1}$) | $\Delta E$ (kJ·mol$^{-1}$) |
|---|---|---|---|---|
| 1 | 1164 | $(1)^{1side}$ | −3590 | 0.0 (+10.3) |
| | | $(2)^{1vex}$ | −3550 | +0.5 (+7.5) |
| | | $(3)^{1cav}$ | −3440 | +1.8 (0.0) |
| 2 | 1166 | $(1)^{2vex}$ | −5890 | 0.0 (+16.5) |
| | | $(2)^{2cav-side}$ | −5700 | +2.3 (0.0) |
| | | $(3)^{2cav}$ | −5680 | +2.5 (+3.4) |
| | | $(4)^{2vex-side}$ | −5650 | +12.9 (+9.5) |
| 3 | 1168 | $(1)^{3cav}$ | −9200 | 0.0 (0.0) |
| | | $(2)^{3vex}$ | −8860 | +4.1 (+10.5) |
| | | $(3)^{3vex-side}$ | −8610 | +7.1 (+12.3) |
| | | $(4)^{3cav-side}$ | −8420 | +9.3 (+0.4) |

In the case of $n$ = 3, the isomer in which the water cyclic trimer, located in the [111] plane adjacent to that of corannulene, with its center approximately on the $C_{5V}$ axis of corannulene, interacts with the concave face of corannulene through three H atoms ($(1)^{3cav}$, see Figure 5), is shown to be particularly stable with our model. Contrary to the smaller clusters ($n$ = 1, 2), it is also the case after relaxation of the cluster after removing the Ar matrix as this conformer remains the most stable one in the gas phase. Next isomer, $(2)^{3vex}$,

in which the cyclic trimer interacts with the convex face of corannulene through two H atoms, is found 340 cm$^{-1}$ higher in energy. The next two isomers in which the water trimer adopts a "linear form" interacting with both one face (convex or concave) and an H atom of corannulene, namely the (3)$^{3vex-side}$ and (4)$^{3cav-side}$ are found respectively 590 and 780 cm$^{-1}$ (7.1 and 9.3 kJ·mol$^{-1}$) higher in energy than the stable (1)$^{3cav}$ isomer.

### 3.1.2. IR Spectra

This section reports the computed IR spectra of all the $(C_{20}H_{10})(H_2O)_n$:Ar isomers for $n$ = 1–3 and the comparison with those of $C_{20}H_{10}$ and of the corresponding $(H_2O)_n$ clusters determined at the same level of theory. These simulations aim at getting insights into the influence of the coordination site (concave, convex or side) and of the geometry of the water cluster (linear or cycle)—in the case of the trimer—on the IR spectrum of the water molecule/cluster and corannulene upon coordination. The values of the positions and the shifts of the bands of interest upon coordination are reported in Tables A2–A4 for the harmonic spectra ($n$ = 1 to 3 respectively), whereas Tables 2–4 report the data for finite-temperature (10 K) spectra. For the latter, the maxima were determined using the full width at half maximum (FWHM). Due to the fluctuations of the simulated IR cross sections and the significant broadening and merge of bands in some cases, we can estimate a human error of ±5 cm$^{-1}$ on the reported wavenumbers. The 10 K dynamic spectra of all the isomers reported in Figures 3–5 are displayed in Figures 6–8.

First of all, we found that, for all clusters, the bands of corannulene were hardly affected upon coordination, and it is in line with the experimental results. This is clearly the case of the $\nu_{CH}$ (C-H stretch) bands, and for the $\delta$(CH) + $\nu$(CC) bands (combination of in plane C-H bending and C-C stretches) that have very weak intensities in the computed spectra. Regarding the $\gamma_{CH}$ (C-H out-of-plane) most modes, their energy can be slightly blue-shifted for the "side" isomers all the more as the cluster size increases. The most intense $\gamma_{CH}$ band is shifted by +5 cm$^{-1}$ for (1)$^{1-side}$, +9 cm$^{-1}$ for (2)$^{2cav-side}$, +8 and +14 cm$^{-1}$ for (3)$^{3vex-side}$ and (4)$^{3cav-side}$ respectively using the harmonic data. This small blue shift was also observed for the $\sigma$ isomer the coronene-water complex [18]. The computed dynamic shifts were found similar to the harmonic ones. Therefore a side isomer may be distinguished from a convex or concave isomer through the position of the $\gamma_{CH}$ band as in the case of coronene [18]. It should be noticed that these shift values are small with respect to the accuracy of the computed absolute positions. However, the description of intermolecular interactions has been carefully benchmarked against wavefunction results, insuring their accuracy [48].

In the following, we focus on the influence of the interaction with corannulene on the IR spectra of $(H_2O)_n$:Ar ($n$ = 1–3).

Regarding the dynamic spectra of the isomers with one water molecule (Figure 6), the first striking feature is the narrowing of the water bands upon coordination. Indeed, in our simulations of $H_2O$:Ar at 10 K, the water molecule vibrates and rotates in the matrix, leading to the broadening of the resonant symmetric and assymmetric stretching modes ($\nu_1$ and $\nu_3$) as well as the bending mode ($\nu_2$). Besides, the stretching modes are significantly redshifted with respect to the harmonic ones, as we reported in our previous work focusing on the IR spectra of the water molecules and clusters inside a rare gas matrix [56]. As can be seen in Tables A2 and 2, the shifts are quite variable depending on the isomer. For the (1)$^{1side}$ isomer, the $\nu_1$ and $\nu_3$ modes are blueshifted upon coordination, the blue shift being more pronounced in the anharmonic spectrum (+38 and +35 vs. +4 and +2 cm$^{-1}$ for the anharmonic and harmonic $\nu_1$ and $\nu_3$ modes respectively). On the opposite, in the case of the (2)$^{1vex}$ isomer, the $\nu_1$ and $\nu_3$ bands are redshifted, the shifts being less pronounced in the dynamic spectra (−36 and −43 cm$^{-1}$ vs. −17 and −1 cm$^{-1}$). The case of (3)$^{1cav}$ is intermediate as the $\nu_1$ and $\nu_3$ modes are redshifted at the harmonic level and become slightly blueshifted in the 10 K dynamic spectra (Figure 6). Regarding the $\nu_2$ mode, it it blueshifted (except in the (1)$^{1side}$ harmonic spectrum) increasing from the side to convex and concave isomers. However, the significant narrowing of the water bands upon coordination may lead to the impossibility to detect the $(C_{20}H_{10})(H_2O)$:Ar clusters in the matrix as they can be hindered by the bands of the water molecule.

For larger clusters, only the dynamic spectra will be discussed. Regarding $(C_{20}H_{10})_{0,1}$ $(H_2O)_2$:Ar, let us just specify that, regarding the 10 K spectrum of the water dimer, the $\nu_1$ and $\nu_3$ modes are redshifted with respect to the harmonic spectra. The shifts obtained for the O-H modes for the H atoms interacting with the Ar atoms of the matrix (the $\nu_1$ and $\nu_3$ most energetic modes, usually designated as $\nu_{1a}$ and $\nu_{3a}$, "a" standing for "acceptor") are the most shifted (both by 48 cm$^{-1}$) whereas the lowest energetic ones ($\nu_{1d}$ and $\nu_{3d}$, "d" standing for "donor") are only shifted by 7 and 9 cm$^{-1}$ respectively. It can be understood as the former interact with the Ar atoms of the matrix and therefore are more sensitive to the loose interaction with Ar expected to lead to anharmonic effects at low temperature. Extremely weak intensity bands also accompany the intense ones, to a lesser extent than for the water monomer because the motion is more hindered by the matrix environment. The water dimer vibrates around its equilibrium position with the possible occurence of the rotation of the H-acceptor water monomer around the OO axis. Interestingly, the water dimer motion in our simulations appears similar to the experimental evidence of the "acceptor switching and rotation of the water dimer around its O-O axis" by Ceponkus et al. [58].

**Table 2.** Positions (cm$^{-1}$) of the IR anharmonic bands in matrix of the corannulene, water monomer and the corresponding $(C_{20}H_{10})(H_2O)$ isomers. For the latter, the shifts of the water modes induced by the adsorption on corannulene are also reported (/). For $H_2O$, the numbers in parenthesis are the positions determined for the longer simulations (see Appendix B and Figure A2). Experimental data are mentionned for comparison. [a] this work, [b] [59] [c] [60].

|  | Mode | This Work | Expt. | (1)$^{1side}$ | (2)$^{1vex}$ | (3)$^{1cav}$ |
|---|---|---|---|---|---|---|
|  | $\gamma$(CH) | 811 | 837 [a] | 816 | 812 | 813 |
|  | $\delta$(CH) | 1171 | 1139 [a] | 1180 | 1175 | 1174 |
|  |  | 1270 |  | 1270 | 1269 | 1269 |
| $C_{20}H_{10}$ | $\delta$(CH) + $\nu$(CC) | 1640 | 1316 [a] | 1639 | 1641 | 1640 |
|  | $\delta$(CH) + $\nu$(CC) | 1849 | 1441 [a] | 1852 | 1849 | 1851 |
|  | $\nu$(CH) | 3013 | 3050 [a] | 2996 | 3015 | 3017 |
| (H$_2$O) | $\nu_1$ | 3756 (3760) | 3638.5 [b], 3639.2 [c] | 3794 /+38 | 3739 /−17 | 3763 /+7 |
|  | $\nu_2$ | 1561 (1563) | 1589 [b], 1589.1 [c] | 1568 /+7 | 1582 /+21 | 1593 /+32 |
|  | $\nu_3$ | 4021 (4021) | 3733.5 [b], 3737.2 [c] | 4056 /+35 | 4020 /−1 | 4028 /+7 |

**Table 3.** Positions (cm$^{-1}$) of the IR anharmonic bands in matrix of the corannulene, water dimer and the corresponding $(C_{20}H_{10})(H_2O)_2$. For the latter, the shifts of the water modes induced by the adsorption on corannulene are also reported (/). Experimental data are mentionned for comparison. [a] this work, [b] [58].

|  | Mode | This Work | Expt. | (1)$^{2vex}$ | (2)$^{2cav-side}$ | (3)$^{2cav}$ | (4)$^{2vex-side}$ |
|---|---|---|---|---|---|---|---|
|  | $\gamma$(CH) | 811 | 837 [a] | 813 | 820 | 814 | 818 |
|  | $\delta$(CH) | 1176 | 1139 [a] | 1176 | 1174 | 1175 | 1178 |
|  |  | 1270 |  | 1269 | 1267 | 1268 | 1261 |
| $C_{20}H_{10}$ | $\delta$(CH) + $\nu$(CC) | 1640 | 1316 [a] | 1641 | 1639 | 1640 | 1639 |
|  | $\delta$(CH) + $\nu$(CC) | 1849 | 1441 [a] | 1850 | 1852 | 1849 | 1852 |
|  | $\nu$(CH) | 3013 | 3050 [a] | 3020 | 3017 | 3022 | 2995 |
| (H$_2$O)$_2$ | $\nu_1$ | 3658 | 3574 [b] | 3630 /−28 | 3593 /−65 | 3624 /−34 | 3627 /−31 |
|  |  | 3760 | 3633 [b] | 3748 /−12 | 3745 /−15 | 3742 /−18 | 3754 /−6 |
|  | $\nu_2$ | 1562 | 1597 [b] | 1568 /+6 | 1570 /+8 | 1585 /+23 | 1569 /+7 |
|  |  | 1587 | 1610 [b] | 1597 /+10 | 1586 /−1 | 1608 /+21 | 1580 /−7 |
|  | $\nu_3$ | 3982 | 3708 [b] | 3926 /−56 | 3956 /−26 | 3924 /−58 | 3982 /0 |
|  |  | 4024 | 3730 [b] | 4031 /+7 | 4013 /−11 | 4008 /−16 | 4028 /+4 |

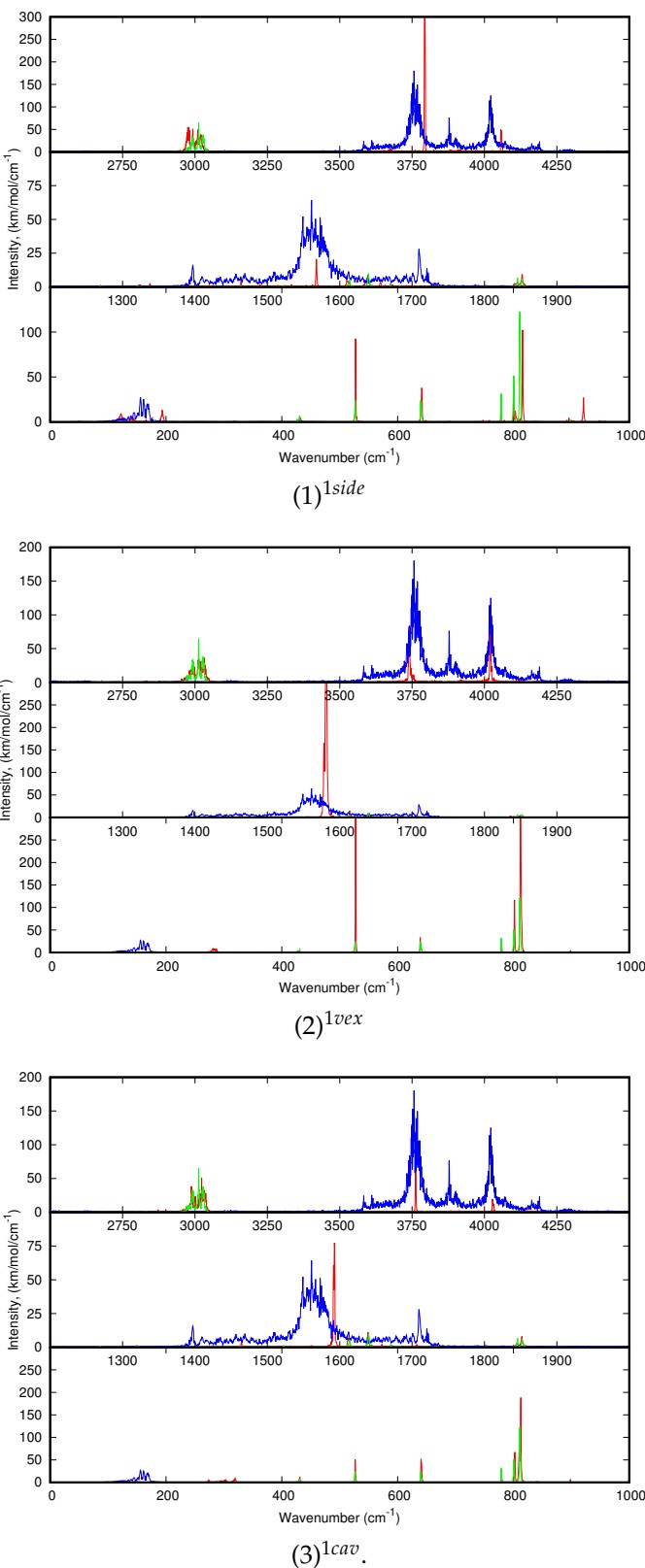

**Figure 6.** IR spectra derived from MD/DFTB-FF simulations at 10 K of the three lowest energy isomers of $(C_{20}H_{10})(H_2O)$:Ar (red), $H_2O$:Ar (blue), and $C_{20}H_{10}$:Ar (green).

**Table 4.** Positions (cm$^{-1}$) of the IR anharmonic bands in matrix of the corannulene, water trimer and the corresponding $(C_{20}H_{10})(H_2O)_3$ isomers. In the case of the presence of broad bands, the maxima were determined using the FWHM, which is indicated in parenthesis. The shifts of the water modes induced by the adsorption on corannulene are also reported (/). * The shift was determined using a maximum position of 1579 cm$^{-1}$ for the $\nu_2$ band of $(H_2O)_3$ estimated considering the structured broad band with a FWHM of 30 cm$^{-1}$. Experimental data are mentionned for comparison. [a] this work, [b] [61].

| | Mode | This Work | Expt. | $(1)^{3cav}$ | $(2)^{3vex}$ | $(3)^{3vex-side}$ | $(4)^{3cav-side}$ |
|---|---|---|---|---|---|---|---|
| | $\gamma(CH)$ | 811 | 837 [a] | 815 | 813 | 819 | 825 |
| | $\delta(CH)$ | 1176 | 1139 [a] | 1175 | 1175 | 1180 | 1176 |
| | | 1270 | | 1269 | 1273 | 1272 | 1271 |
| $C_{20}H_{10}$ | $\delta(CH) + \nu(CC)$ | 1640 | 1316 [a] | 1639 | 1639 | 1639 | 1638 |
| | $\delta(CH) + \nu(CC)$ | 1849 | 1441 [a] | 1851 | 1849 | 1851 | 1852 |
| | $\nu(CH)$ | 3013 | 3050 [a] | 3008 | 3017 | 3012 | 3005 |
| | | | | | | 3554 (12) / −42 | 3537 (11) / −59 |
| | $\nu_1$ | 3596 (50) | 3514 [b] | 3591 (12) / −5 | 3605 (49) / +9 | 3595 (10) / −1 | 3584 (10) / −12 |
| | | | | | | 3721 (10) / +125 | 3741 (8) / +145 |
| | | 1566 | | 1566 (0) | 1548 / −18 | | 1577 / +11 |
| $(H_2O)_3$ | $\nu_2$ | 1570 | 1601 [b] | 1571 / +1 | 1581 / +11 | 1586 (20) / +7 * | 1592 / +22 |
| | | 1591 | | 1599 / +8 | 1600 / +9 | | 1594 / +3 |
| | | | | | | 3944 / −4 | 3940 / −8 |
| | $\nu_3$ | 3948 (26) | 3699 [b] | 3902 (13) / −46 | 3950 (76) / +3 | 3960 / +12 | 3954 / +6 |
| | | | | | | 4012 / +64 | 4001 / +53 |

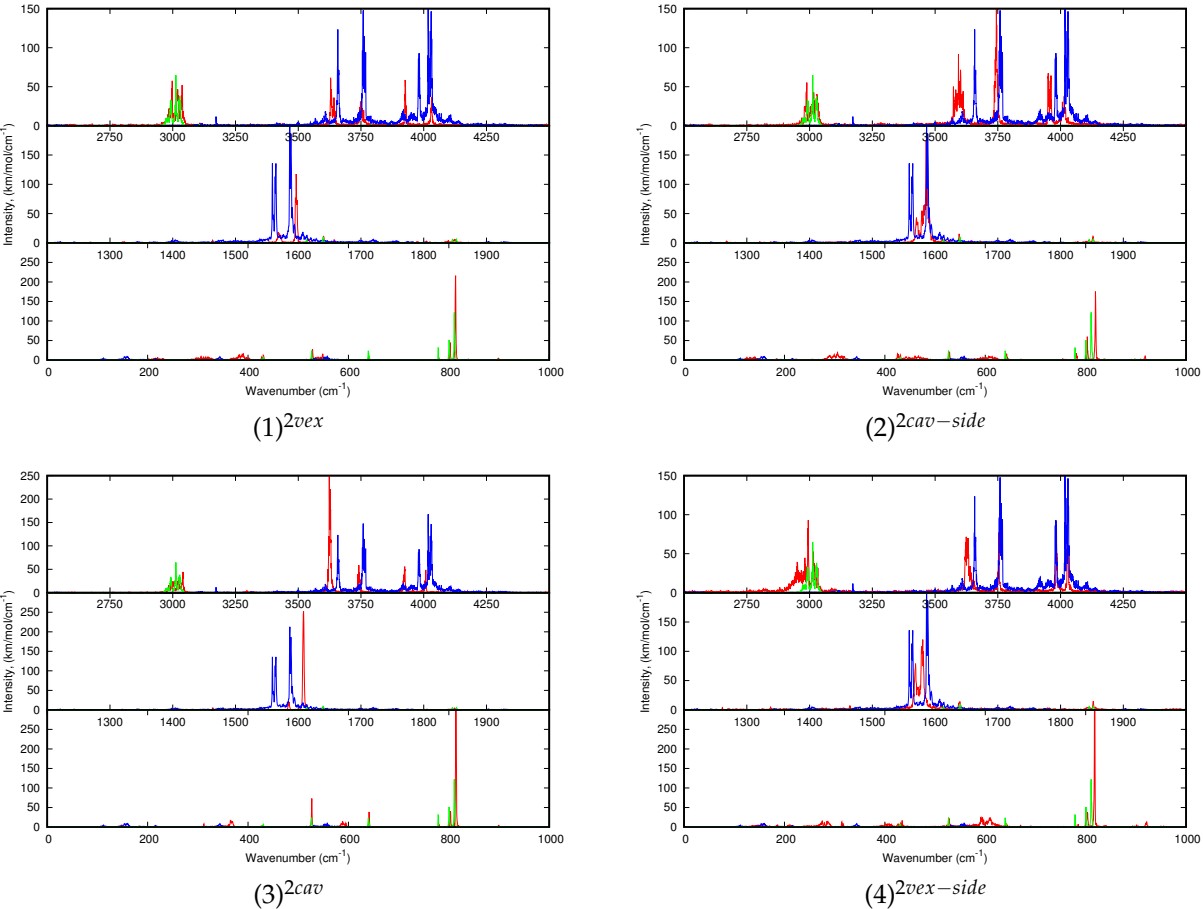

(1)$^{2vex}$      (2)$^{2cav-side}$

(3)$^{2cav}$      (4)$^{2vex-side}$

**Figure 7.** IR spectra derived from MD/DFTB-FF simulations at 10 K of the four lowest energy isomers of $(C_{20}H_{10})(H_2O)_2$:Ar (red), $(H_2O)_2$:Ar (blue), and $C_{20}H_{10}$:Ar (green).

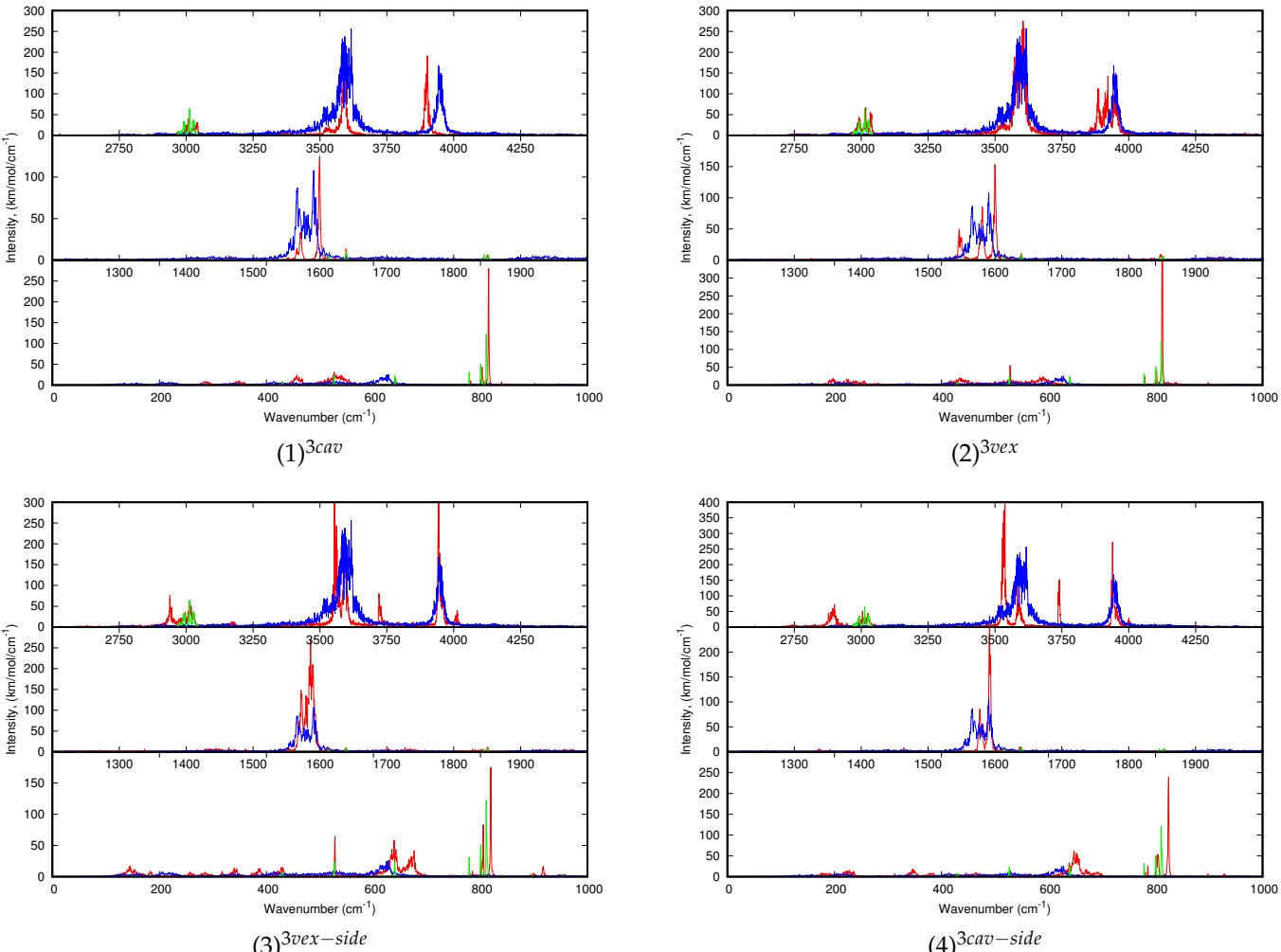

**Figure 8.** IR spectra derived from MD/DFTB-FF simulations at 10 K of the four lowest energy isomers of $(C_{20}H_{10})(H_2O)_3$:Ar (red), $(H_2O)_3$:Ar (blue), and $C_{20}H_{10}$:Ar (green).

Regarding the spectra of all $C_{20}H_{10})(H_2O)_2$:Ar isomers reported Figure 7, the differences in the shifts of the bands with respect to those of the bare water dimer might help identifying the type of isomer (cav, vex , cav-side or vex-side) formed in the experiments. Upon interaction with corannulene, the $(H_2O)_2$ $\nu_1$ modes are globally redshifted and the shifts values are smaller in the 10 K spectra than in the harmonic ones. The shifts values are dependent on the isomer being smaller for the $(1)^{2vex}$ and $(4)^{2vex-side}$ isomers (see Table 3). The $\nu_3$ bands are redshifted for $(2)^{2cav-side}$ and $(3)^{2cav}$, but is it not the case for the two other isomers for which the higher energy $\nu_3$ band is blueshifted. Finally, a difference between the shifts of the $\nu_2$ bands between the different isomers is noticeable. For instance, those for $(1)^{2vex}$ and $(3)^{2cav}$ are both blueshifted, the blue shift being larger for the latter.

Finally, the dynamic spectra of $(C_{20}H_{10})_{0,1}(H_2O)_{0,3}$:Ar are reported in Figure 8. Regarding the water modes of $(H_2O)_3$, each type of mode leads to three broad bands more or less structured centered at 3596, 1569 and 3948 cm$^{-1}$ for $\nu_1$ to $\nu_3$. During the simulation, the water trimer vibrates around its equilibrium position with occasional flipping of the O-H bonds- not involved in intermolecular O–H bonds- from one side of the plane of the OOO cycle to the other side. As in the case of one and two water molecules, the water IR bands become narrower due to a hindrance of their motion upon interaction with corannulene, although it is not as relevant in the case of $(2)^{3vex}$ where the water trimer globally maintains its structure. It is all the more verified for the "cav" isomers $(1)^{3cav}$ and $(4)^{3cav-side}$, reflecting

stronger corannulene-water interactions than for the "vex" isomers. A significant difference is noticeable between the "side" isomers and the others. In the former, a band can be found at ∼3720 and 3740 cm$^{-1}$ , that was assigned to the O-H stretching mode for the bond interacting with the PAH [51] . In contrast with the other stoichiometries, the stretching modes do not appear systematically reshifted. The $\nu_1$ mode is hardly redshifted in the case of the most stable $(1)^{3cav}$ isomers whereas is it slightly blueshifted in the $(2)^{3vex}$ spectrum. The $\nu_3$ mode is redshifted for $(1)^{3cav}$ but leads to mostly blueshifted bands for the three other isomers. The $\nu_2$ modes are also globally blueshifted for all isomers, the amplitude of the shifts being dependent on the isomer. As in the case of the other stoichiometries, the differences in the spectra may allow us to distinguish specific isomers formed in the experiment.

### 3.2. Experiments

In this section, we present the FTIR spectra of water, corannulene, and water:corannulene mixtures recorded in argon matrices at 10 K, where water is present in the form of monomers, dimers, trimers and larger aggregates, depending on the $H_2O$:Ar ratio. The data will be compared to the computational results presented in the above section and discussed in the next section.

The IR spectra of water in solid argon have been previously studied in detail by several authors (see, for example, refs. [59–67]). Our experimental spectra agree with literature spectra and our assignments are based on them. The assignments of water monomers and water aggregates formed in our deposits will allow us to identify perturbed bands (shifts) and/or additional bands induced by the presence of corannulene. Figure A3 (a reproduction of Figure 3 in ref. [18]) presents the FTIR spectra of different $H_2O$:Ar deposits at 10 K with increasing concentrations of water: (a) $H_2O$:Ar = 1:500, where only monomers and dimers are present; (b) $H_2O$:Ar = 1:50, where larger aggregates (from trimers to hexamers) appear; and (c) $H_2O$:Ar = 1:25, where a mixture of water aggregates and water ice (ASW) is observed. These spectra have been discussed in detail in our previous paper [18] and Table A5 lists the main band positions of water aggregates $(H_2O)_n$, with $n$ = 1–6, as well as those of some water complexes $H_2O$:$CO_2$, $H_2O$:$N_2$ and $(H_2O)_2$:$N_2$.

Although corannulene and coronene are both aromatic, they present marked differences in structure and reactivity [20]. Due to its C5v symmetry, and its bowl-like non-planar structure, the corannulene molecule exhibits a richer IR spectrum than coronene, with more active modes and larger intensities. Table A6 lists the vibrational modes observed and calculated in this work, compared to experimental data from the NASA Ames PAH IR spectral database [68–70] and previous study by Rouillé et al. [39]. A summary of the main experimental bands is reported in Tables 2–4 where the frequencies are compared to the IR anharmonic band positions calculated in this work.

Figure 9 presents the FTIR spectra of corannulene in Ar and in $H_2O$:Ar (1:50) in two spectral regions of interest: 2000–400 cm$^{-1}$, where we can observe the main corannulene bands plus the deformation modes $\nu_2$ of water; and 4000–2800 cm$^{-1}$, where we can observe all OH stretching modes of water monomers and aggregates plus the CH stretching bands of corannulene. Despite several attempts, it was impossible to obtain a pure corannulene:Ar spectrum without any traces of water; a small amount of water monomer (lines at 3756 and 3778 cm$^{-1}$) and trace of $(H_2O)_2$ (line at 3708 cm$^{-1}$) are always present in the experimental corannulene:Ar spectra. No change is observed in water monomer lines, nor in corannulene lines in this spectrum, so we consider that the trace amount of water in the matrix is not interacting with the corannulene.

If we now consider depositions with intentionally added water $H_2O$:Ar 1:50 (Figure 9, red traces labelled c), we can see that, apart from a minor broadening, the vibrational bands of corannulene remain almost unchanged in frequency compared to corannulene in argon. Contrary to what was observed in the case of coronene [17], where the most intense out-of-plane $\gamma$(CH) mode was blue-shifted by up to 12 cm$^{-1}$ upon interaction with water aggregates, no or very small blue shifts (a few cm$^{-1}$ i.e., within the spectral resolution) are observed for the $\gamma$(CH) mode of corannulene at 837 cm$^{-1}$. The main changes in the FTIR

spectrum appear in the OH stretching modes of water, especially in the bands assigned to water dimers and trimers. Small blue shifts and broadenings are observed for the bands located at 3700 (trimer), 3706 (dimer) and 3712 (monomer) cm$^{-1}$. No change is detected below 3500 cm$^{-1}$, in the region of the stretching modes of $(H_2O)_n$ with $n = 4$–6.

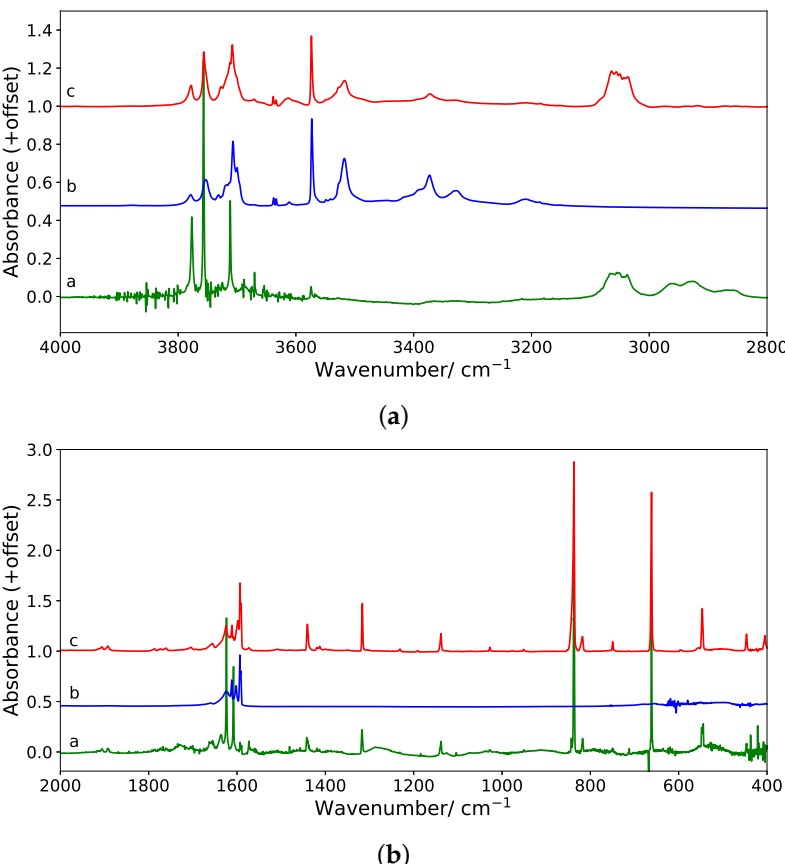

**Figure 9.** Experimental spectra measured in the wavelength ranges (**a**) 4000–2800 cm$^{-1}$ and (**b**) 2000–400 cm$^{-1}$. Spectra in both panels are as follows: a—Corannulene in argon (green trace), b—water in argon (blue trace), and c—corannulene and water in argon (red trace).

Figure 10 presents the difference spectrum between the corannulene:$H_2O$:Ar deposition minus the $H_2O$:Ar deposition at 10 K in the 3800–3200 cm$^{-1}$ and 1800–1500 cm$^{-1}$ ranges, where stretching and bending water modes are, respectively, located. A new band appears at 3727 cm$^{-1}$, as well as two broad, weak bands at 3603 and 3613 cm$^{-1}$. These features are in proximity to a weak band in the $H_2O$:Ar spectrum assigned to a symmetric stretching mode of the water trimer at 3611 cm$^{-1}$. A weak band is also observed at 3536 cm$^{-1}$, in the region of the symmetric stretching modes of water dimers (3574 cm$^{-1}$) and trimers (3518 cm$^{-1}$). Additionally, a new band clearly appears at 1598 cm$^{-1}$, in the HOH bending region of water dimers at 1593 cm$^{-1}$ and water trimers at 1602 cm$^{-1}$. We can thus conclude that molecular interaction between water dimers and/or trimers with corannulene are observed in the FTIR spectra.

Energy was introduced into the corannulene:$H_2O$:Ar samples via photons. Low energy UV irradiations of our deposition did not induce any change in the spectrum, contrary to what was previously observed for coronene:$H_2O$:Ar matrices at 10 K, where the interaction of coronene with water aggregates $(H_2O)_n$ with n up to 3 led to the formation of oxygenated products [17].

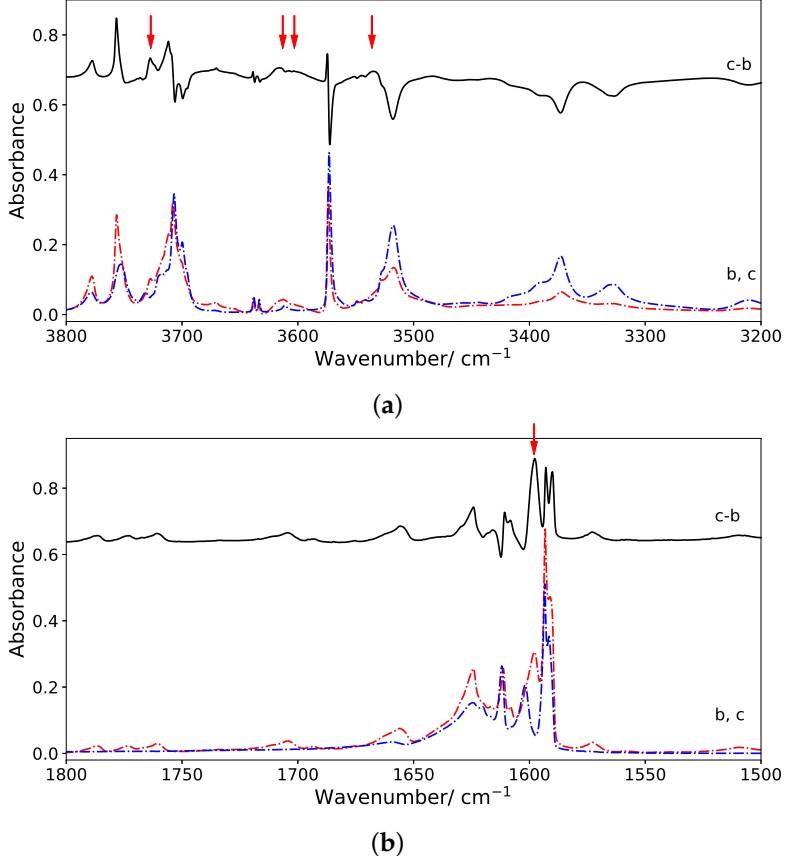

**Figure 10.** Experimental difference spectra in the wavelength ranges (**a**) 3800–3200 cm$^{-1}$ and (**b**) 1800–1500 cm$^{-1}$. Spectra b (water, blue) and c (corannulene plus water, red) of Figure 9 are reproduced in dot-dashed lines, while the solid black line represents their difference spectrum (i.e., spectrum c minus spectrum b). New bands discussed in the text are indicated with red arrows at the following frequencies: 3727, 3613, 3603, 3536, and 1598 cm$^{-1}$.

## 4. Discussion

In this section, we discuss the assignment of the experimental spectra in the light of the calculated finite-temperature spectra (Tables 2–4 for corannulene in interaction with, respectively, water monomers, dimers and trimers). As the absolute positions of the calculated frequencies are not strictly accurate, we will use the calculated shifts to identify the $(C_{10}H_{20})(H_2O)_n$ complexes formed in our experiments. Indeed, regarding the computed absolute positions, we are aware that they are not accurate. It is due to several reasons: (i) the DFTB method is a DFT-based approximate method (truncated hamiltonian, minimal valence basis set…). However, the DFTB/FF method used in the present work, quite unique to our knowledge, is mandatory to be able to perform on-the-fly BOMD simulations taking explicitly into account all atoms (including the atoms of the rare gas matrix). The possible presence of defaults is not taken into account. (ii) 0 K vibrational anharmonicities and nuclear quantum effects, expected to be significant in the case of water clusters, are not taken into account, and it is a major drawback of our computational studies. However, given the size of the systems and the presence of highly symmetric corannulene, an approach such as vibrational perturbational theory using a wavefunction method to describe the electronic structure in conjunction with a large basis set, which would allow to obtain accurate 0 K IR spectra for such systems, is prohibited. In our approach however, as specified in Section 3.1.2, intermolecular interactions are expected to be better described than intramolecular bonds as they have been carefully benchmarked against wavefunction results [48].

The new band observed at 3727 cm$^{-1}$ may be due to a blue shift of 15 cm$^{-1}$ of the $\nu_3$ mode of the water monomer (3712 cm$^{-1}$) in $(C_{10}H_{20})(H_2O)$ "side" or "cav" geometries

(see Table 2), or to a blue shift of 21 cm$^{-1}$ of the $\nu_3$ mode of the water dimer (3706 cm$^{-1}$) in $(C_{10}H_{20})(H_2O)_2^{vex}$ (see Table 3). These two values being located between those calculated for these two complexes (7 to 35 cm$^{-1}$), it is impossible to conclude at this stage. However, it seems unlikely that $(C_{10}H_{20})(H_2O)$ with "side" structure has been formed, as no large red shift has been observed on the $\nu$(CH) of corannulene, whereas $(C_{10}H_{20})(H_2O)$ in the "cav" form, calculated to induce only a small blue shift in the $\nu$(CH) modes, could be a possible candidate (see Table 2).

If we now consider the new bands observed at 3603 and 3613 cm$^{-1}$, they may be red-shifted from the $\nu_3$ modes of the water dimer (3706 cm$^{-1}$, shift of 93 cm$^{-1}$) or trimer (3700 cm$^{-1}$, shift of 87 cm$^{-1}$) , or blueshifted from the $\nu_1$ mode of the water dimer (3574 cm$^{-1}$, shifts of 29 and 39 cm$^{-1}$). From Tables 3 and 4, it appears that no blue shifts are calculated for the $\nu_1$ modes of any $(C_{10}H_{20})(H_2O)_2$ structures nor for $(C_{10}H_{20})(H_2O)_3^{cav}$, whereas large red shifts are expected for the $\nu_3$ modes of these complexes (from 46 to 58 cm$^{-1}$). So, the more likely assignment for these two new bands is to $(C_{10}H_{20})(H_2O)_2$ "vex" or "cav" forms and to $(C_{10}H_{20})(H_2O)_3^{cav}$. For the same reasons, the weak band observed at 3536 cm$^{-1}$ must be red shifted from the $\nu_1$ mode of the water dimer (3574 cm$^{-1}$, shift 38 cm$^{-1}$) in the $(C_{10}H_{20})(H_2O)_2$ "vex" or "cav" forms, as only a small blue shift (5 cm$^{-1}$, see Table 4) is expected from the $\nu_1$ mode of the water trimer observed at 3518 cm$^{-1}$.

Finally, the new band at 1598 cm$^{-1}$ could be due to a blue shift of 6 or 5 cm$^{-1}$ of the $\nu_2$ mode of, respectively, the water monomer observed at 1592 cm$^{-1}$ or of the dimer at 1593 cm$^{-1}$, or could be due to a red shift of 4 cm$^{-1}$ of the $\nu_2$ mode of the water trimer at 1602 cm$^{-1}$. Comparison with calculated shifts in Tables 3 and 4 shows that only blue shifts are expected for all $\nu_2$ modes in all complexes, whatever the stoichiometry and structures. However, this blue shift is calculated to be 7 cm$^{-1}$ in $(C_{10}H_{20})(H_2O)^{side}$, but more than 30 cm$^{-1}$ in $(C_{10}H_{20})(H_2O)^{cav}$. As we have already eliminated $(C_{10}H_{20})(H_2O)^{side}$ due to the absence of shifts in the $\nu$(CH) bands of corannulene, we thus assign this new band to $(C_{10}H_{20})(H_2O)_2^{vex}$.

From the analysis of our experimental spectra—which is complicated by the presence of excess corannulene and water aggregates ($n$ = 1 to 6)—and thanks to our calculations, we can conclude that: (1) no $(C_{10}H_{20})(H_2O)$ complex has been detected in our experimental conditions. This can be due to the effect of argon environment, or to a better reactivity of corannulene with small aggregates (dimer, trimer) compared to the water monomer, (2) the $(C_{10}H_{20})(H_2O)_3$ complex with the water trimer interacting with the concave face of corannulene has been identified, which is in line with the fact that it is the most stable calculated structure, and (3) the $(C_{10}H_{20})(H_2O)_2$ complex is also formed, but it is impossible to firmly characterize its structure, which may be "vex" or "cav". We have not seen any experimental evidence for the formation of complexes with larger aggregates ($n$ = 4 to 6) on the difference spectra. Even if their formation cannot be completely excluded, it is hardly probable that they could be formed, because the water tetramer and larger cluster spectra are quite different from those of water dimer and trimer and located lower in energy.

## 5. Conclusions

Our study has shown that the weak PAH-water complexes formed in rare gas matrices may be different from those obtained in the gas phase. Our prior computations have shown that embedding in the matrix stabilises $\sigma$ isomers in the case of planar aromatics [18], while for the clusters involving the non-planar corannulene studied in this work, $\pi$ isomers remain favored in the cases of the water dimer and trimer. Such geometrical differences may account for the differences in photo-reactivity of these systems: indeed, pyrene and coronene react with water molecules to form alcohols and quinones under low energy irradiation [13,14] while corannulene does not, even when embedded in ASW (private comm. from J. Mascetti). The photoreactivity in planar PAHs could occur through the population of low energy charge transfer PAH$^+$-H$_2$O$^-$ electronic excited states that are present when the water molecules interact through their oxygen atoms with the H atoms of the PAH [17,19]. Therefore, from the modeling reported in the present work, the most stable form of $(C_{20}H_{10})(H_2O)_3$ in an argon matrix does not appear favorable to the presence

of such low energy excited states. The investigation of the electronic excited states of $(C_{20}H_{10})(H_2O)_n$:Ar systems ($n = 1–3$) will be the object of future work. Our studies illustrate the dependence of the mutual orientation of the reactants on the nature of bimolecular photo-redox reactions as evoked in a recent review [71]. The differentiation between flat and curved PAHs may be of importance in UV absorption and electron transfer reactions in PAH:water molecular systems. From an astrophysical point of view, even if it is still premature to project these results directly to PAHs of astro-relevant sizes embedded in water ice, there is probably no difference in the photoreactivity between flat and curved PAHs when using high energy UV photons that lead to PAH ionization, but the influence of geometry should be taken into account when using low energy UV photons that induce charge transfer reactions.

**Author Contributions:** All authors contributed to the work. H.L. and A.S. performed calculations; C.A. and J.M. ran experiments. All authors have discussed the results. Draft preparation and writing were done by H.L., A.S., J.A.N. and J.M. All authors have read and agreed to the submitted version of the manuscript.

**Funding:** This research was funded by the French National Agency for Research (ANR PARCS project ANR-13-BS008-0005) and supported by the Programme National Physique et Chimie du Milieu Insterstellaire (PCMI) of CNRS/INSU with INC/INP co-funded by CEA and CNES, and by the French research network EMIE (Edifices Moléculaires Isolés et Environnés, GDR 3533).

**Institutional Review Board Statement:** Not applicable.

**Informed Consent Statement:** Not applicable.

**Data Availability Statement:** The data published in the present paper are made public in the archived datasets on the Zenodo website (https://doi.org/10.5281/zenodo.6368179) published on 18 March 2022.

**Acknowledgments:** H.L. and A.S. thank the computing mesocenter CALMIP (UMS CNRS 3667) for generous allocation of computer resources (p17002). J.M. and C.A. acknowledge the participation of internship students Maxime Gerbeaud-Lassau and Francisco Salvador in the experimental part of this study.

**Conflicts of Interest:** The authors declare no conflict of interest.

## Abbreviations

| | |
|---|---|
| AIBs | aromatic infrared bands |
| ASW | amorphous solid water |
| BOMD | Born-Oppenheimer molecular dynamics |
| DFTB | density functional based tight binding |
| FF | force field |
| IR | infrared |
| FTIR | Fourier transform infrared |
| ISM | interstellar medium |
| PAH | polycyclic aromatic hydrocarbon |
| UV | ultraviolet |

## Appendix A. Energetics: Insertion of Corannulene in the Matrix

Different insertion planes for corannulene inside the Ar matrix were tried. The substitution energies are reported in Table A1.

**Table A1.** Substitution energies for the insertion of the corannulene in different cristallographic planes.

| Plane | Number of Ar Removed | $E_{sub}$ (cm$^{-1}$) |
|---|---|---|
| (100) | 9 | 1400 |
| (110) | 5 | 3910 |
| (111) | 7 | $-2470$ |
| Cavity | 13 | 3080 |

## Appendix B. Convergence of IR Dynamic Spectra

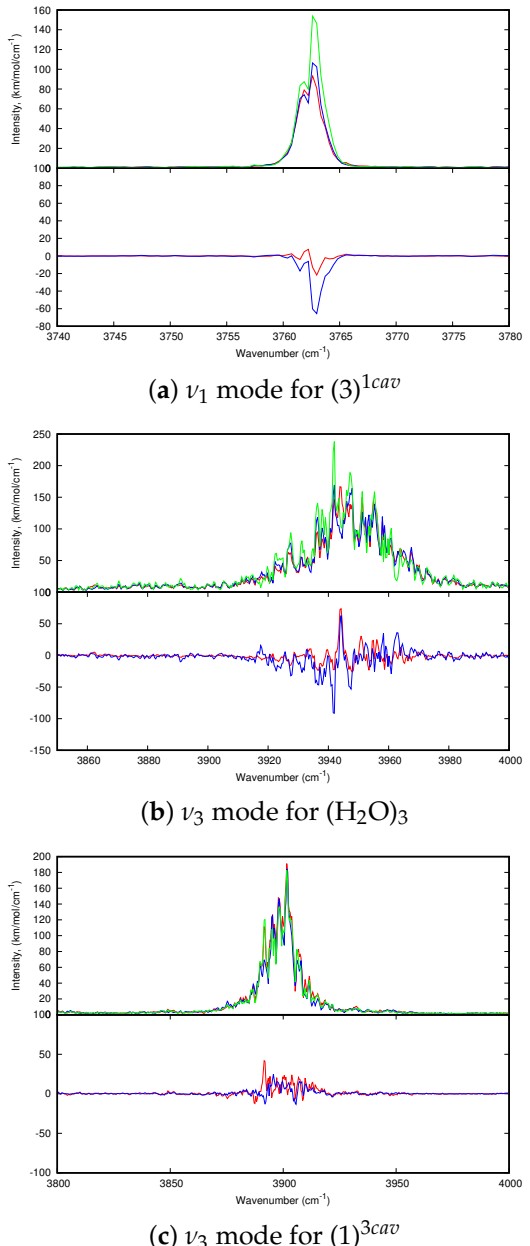

(**a**) $\nu_1$ mode for $(3)^{1cav}$

(**b**) $\nu_3$ mode for $(H_2O)_3$

(**c**) $\nu_3$ mode for $(1)^{3cav}$

**Figure A1.** IR dynamic spectra at 10 K obtained following the procedure described in Section 2.1 for $(3)^{1cav}$ (**a**), $(H_2O)_3$ (**b**) and $(1)^{3cav}$ (**c**). Zooms on the $\nu_1$ (**a**) and $\nu_3$ (**b,c**) modes are presented as an illustration of the convergence. In the upper panel of each plot we present the spectra averaged over $7 \times 100$ ps (red), $5 \times 100$ ps (blue) and $3 \times 100$ ps (green). In the lower panel of each plot we show the difference spectra between $7 \times 100$ ps and $5 \times 100$ ps dynamics (red) and between $7 \times 100$ ps and $3 \times 100$ ps MDs (blue). The resolution of all spectra is 0.4 cm$^{-1}$.

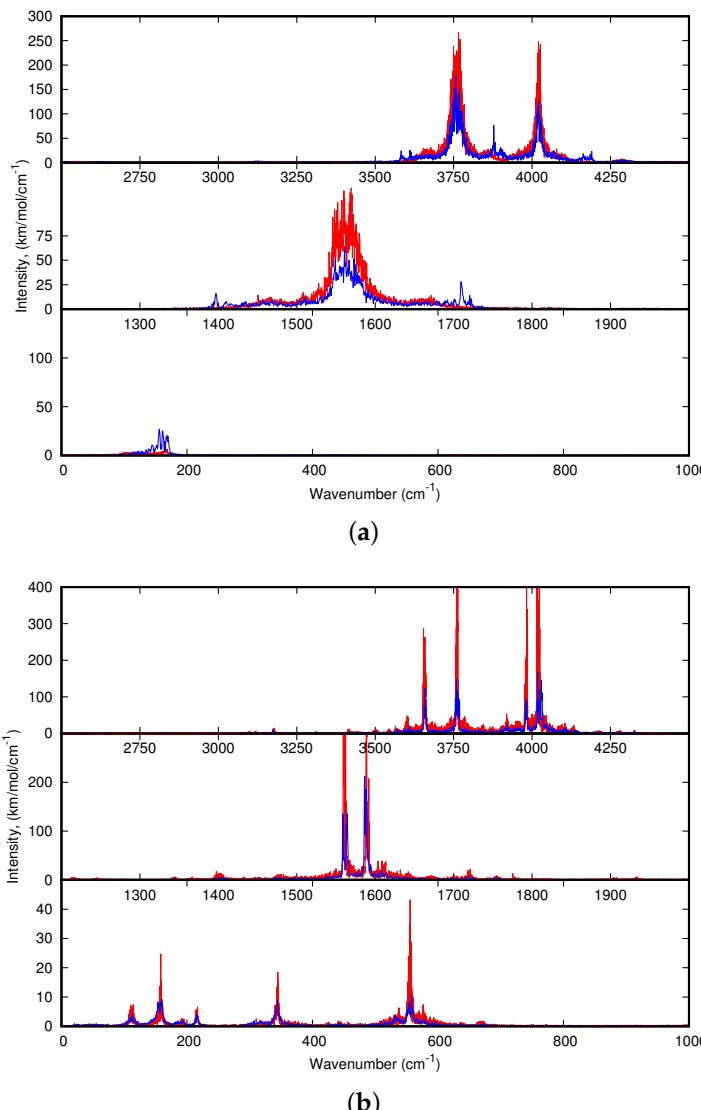

**Figure A2.** IR dynamic spectra at 10 K for $H_2O$:Ar (**a**) and $(H_2O)_2$:Ar (**b**). The spectra of Figures 6 and 7, resulting from $7 \times 100$ ps MDs, are reported in blue. Those resulting from longer simulations, $7 \times 500$ ps for $H_2O$:Ar and $3 \times 500$ ps for $(H_2O)_2$:Ar, are reported in red.

## Appendix C. Harmonic Data

**Table A2.** Positions ($cm^{-1}$) of the IR harmonic bands in matrix of the corannulene, water monomer and the corresponding $(C_{20}H_{10})(H_2O)_1$ isomers. Harmonic IR intensities are reported in parenthesis, as well as shift values (/). Experimental data are also reported for comparison. [a] this work, [b] [59], [c] [60].

| | Mode | This Work | Expt. | (1)[1side] | (2)[1vex] | (3)[1cav] |
|---|---|---|---|---|---|---|
| | $\gamma$(CH) | 812 (147) | 837 [a] | 817 (135) | 814 (149) | 813 (153) |
| | $\delta$(CH) | 1175 (2) | 1139 [a] | 1178 (3) | 1177 (3) | 1176 (3) |
| | | 1271 (<1) | | 1271(<1) | 1271(<1) | 1271 (<1) |
| $C_{20}H_{10}$ | $\delta$(CH) + $\nu$(CC) | 1646 (6) | 1316 [a] | 1644 (6) | 1646 (3) | 1645 (4) |
| | $\delta$(CH) + $\nu$(CC) | 1857 (3) | 1441 [a] | 1856 (3) | 1855 (3) | 1856 (2) |
| | $\nu$(CH) | 2999 * (60) | 3050 [a] | 2999 (62) | 3002 (55) | 3007 (45) |
| | $\nu_1$ | 3807 (39) | 3638.5 [b], 3639.2 [c] | 3811 (138) /+4 | 3771 (52) /−36 | 3765 (57) /−42 |
| $(H_2O)$ | $\nu_2$ | 1571 (140) | 1589 [b], 1589.1 [c] | 1568 (289) /− 3 | 1588 (241) /+17 | 1594 (178) /+23 |
| | $\nu_3$ | 4081 (33) | 3733.5 [b], 3737.2 [c] | 4083 (83) /+2 | 4038 (21) /−43 | 4030 (10)/−51 |

**Table A3.** Positions (cm$^{-1}$) of the IR harmonic bands in matrix of the corannulene, water dimer and the corresponding $(C_{20}H_{10})(H_2O)_2$ isomers. The shifts of the water modes upon interaction with corannulene are also reported (/). Experimental data are mentionned for comparison. [a] this work, [b] [58].

| | Mode | This work | Expt. | (1)$^{2vex}$ | (2)$^{2cav-side}$ | (3)$^{2cav}$ | (4)$^{2vex-side}$ |
|---|---|---|---|---|---|---|---|
| $C_{20}H_{10}$ | $\gamma$(CH) | 812 | 837 [a] | 815 | 821 | 816 | 819 |
| | $\delta$(CH) | 1175 | 1139 [a] | 1174 | 1173 | 1175 | 1179 |
| | | 1271 | | 1271 | 1271 | 1270 | 1271 |
| | $\delta$(CH) + $\nu$(CC) | 1646 | 1316 [a] | 1646 | 1644 | 1645 | 1644 |
| | $\delta$(CH) + $\nu$(CC) | 1857 | 1441 [a] | 1853 | 1855 | 1854 | 1856 |
| | $\nu$(CH) | 2999 | 3050 [a] | 3020 | 3022 | 3028 | 3019 |
| $(H_2O)_2$ | $\nu_1$ | 3665 | 3574 [b] | 3628 / −37 | 3585 / −80 | 3639 / −26 | 3613 / −52 |
| | | 3802 | 3633 [b] | 3745 / −57 | 3757 / −45 | 3757 / −45 | 3752 / −50 |
| | $\nu_2$ | 1568 | 1597 [b] | 1566 / −2 | 1573 / +5 | 1586 / +18 | 1570 / +2 |
| | | 1593 | 1610 [b] | 1597 / +4 | 1593 (0) | 1609 / +16 | 1581 / −12 |
| | $\nu_3$ | 3991 | 3708 [b] | 3925 / −66 | 3962 / −29 | 3941 / −50 | 3983 / −8 |
| | | 4072 | 3730 [b] | 4029 / −43 | 4018 / −54 | 4021 / −51 | 4028 / −44 |

**Table A4.** Positions (cm$^{-1}$) of the IR harmonic bands in matrix of the corannulene, water trimer and the corresponding $(C_{20}H_{10})(H_2O)_3$. The shifts of the water modes upon interaction with corannulene are also reported (/). Experimental data are mentionned for comparison. [a] this work, [b] [61].

| | Mode | This Work | Expt. | (1)$^{3cav}$ | (2)$^{3vex}$ | (3)$^{3vex-side}$ | (4)$^{3cav-side}$ |
|---|---|---|---|---|---|---|---|
| $C_{20}H_{10}$ | $\gamma$(CH) | 812 | 837 [a] | 816 | 814 | 820 | 826 |
| | $\delta$(CH) | 1175 | 1139 [a] | 1174 | 1174 | 1178 | 1177 |
| | | 1271 | | 1269 | 1271 | 1271 | 1273 |
| | $\delta$(CH) + $\nu$(CC) | 1646 | 1316 [a] | 1645 | 1645 | 1644 | 1642 |
| | $\delta$(CH) + $\nu$(CC) | 1857 | 1441 [a] | 1854 | 1853 | 1855 | 1856 |
| | $\nu$(CH) | 2999 | 3050 [a] | 3013 | 3023 | 3021 (+2933) | 3011 (+2888) |
| $(H_2O)_3$ | $\nu_1$ | 3540 | | 3514 / −26 | 3518 / −22 | 3563 / +23 | 3536 / −4 |
| | | 3597 | 3514 [b] | 3580 / −17 | 3576 / −21 | 3602 / +5 | 3595 / −2 |
| | | 3621 | | 3581 / −40 | 3599 / −22 | 3737 / +116 | 3748 / +127 |
| | $\nu_2$ | 1568 | | 1564 / −4 | 1546 / −22 | 1579 / +11 | 1577 / +9 |
| | | 1575 | 1601 [b] | 1569 / −6 | 1580 / +5 | 1587 / +12 | 1581 / +6 |
| | | 1595 | | 1600 / +5 | 1602 / +7 | 1592 / −3 | 1595 (0) |
| | $\nu_3$ | 3960 | | 3900 / −60 | 3894 / −66 | 3959 / −1 | 3951 / −9 |
| | | 3964 | 3699 [b] | 3904 / −60 | 3925 / −39 | 3972 / +8 | 3965 / +1 |
| | | 3971 | | 3913 / −58 | 3959 / −12 | 4017 / +46 | 4008 / +37 |

## Appendix D. Ftir Bands of Water Monomers, Dimers, Trimers and Larger Aggregates in Argon Matrices at 10 K

**Table A5.** From ref. [18] Band positions, in cm$^{-1}$, of $(H_2O)_n$ water species with $n$ = 1 to 6. Attributions are based on previous works [59–62,64–67].

| $n$ | Rovibrational Mode | Our Results (cm$^{-1}$) | Exp Values (cm$^{-1}$) from Litt. | Refs. |
|---|---|---|---|---|
| 1 | $\nu_2 + \nu_3$ | 5346.1, 5325.5 5293.2, 5280.5 | 5345.9, 5325.4 5295.4, 5280.6 | [59,60,62] |
| | $\nu_3$ | 3776.9, 3756.5 3724.8, 3711.4 | 3776.4, 3756.6 3724.9, 3711.3 | |
| | $\nu_1$ | 3669.9, 3653.6 | 3669.7, 3653.5 | |
| | $\nu_2$ RTC | 1662.5, 1657.7 | 1661.4, 1657.2 | |
| | $\nu_2$ | 1636.3, 1624.2 1608.1, 1573.3 1556.9 | 1636.5, 1623.8 1607.9, 1573.1 1556.7 | |
| | $\nu_2$ NRM | 1590.7 | 1589.2 | |

**Table A5.** *Cont.*

| *n* | Rovibrational Mode | Our Results ($cm^{-1}$) | Exp Values ($cm^{-1}$) from Litt. | Refs. |
|---|---|---|---|---|
| 2 | $\nu_3$ PD | 3707.9 | 3708 | |
| | $\nu_3$ PA | 3736.1 | 3737.8 | |
| | $\nu_1$ PA | 3633.1 | 3633.1 | |
| | $\nu_1$ PD | 3573.8 | 3574 | |
| | $\nu_2$ PD | 1611.9 | 1610.6 | |
| | $\nu_2$ PA | 1593.2 | 1593.1 | |
| 3 | $\nu_3$ PD | 3699.8 | 3700 | |
| | $\nu_1$ PA | 3611.1 | 3612 | |
| | $\nu_1$ PD | 3527.9 | 3528 | [59,61,64,65] |
| | $\nu_1$ PD | 3517.4 | 3516 | |
| | $\nu_2$ PA | 1601.9 | 1602 | |
| 4 | | 3372.9 | 3374 | |
| 5 | | 3326.7 | 3327 | |
| 6 | | 3541.3 | 3540 | |
| | | 3444.4 | 3445 | |
| | | 3392.2 | 3391 | [64,67] |
| | | 3210.3 | 3212 | |
| | | 3186.2 | 3195.2 | |
| | | 3151.5 | 3150 | [62] |
| $H_2O$-$CO_2$ | $\nu_1$ | 3638.4 | 3638 | [60] |
| | $\nu_2$ | 1589.5 | 1589.5 | |
| $H_2O$-$N_2$ | $\nu_3$ | 3730.3 | 3729.6 | |
| | $\nu_1$ | 3640.2 | 3640.2 | [66] |
| $(H_2O)_2$-$N_2$ | $\nu_1$ PD | 3566.8 | 3566.6 | |
| | $\nu_1$ PD | 3563.1 | 3563.5 | |

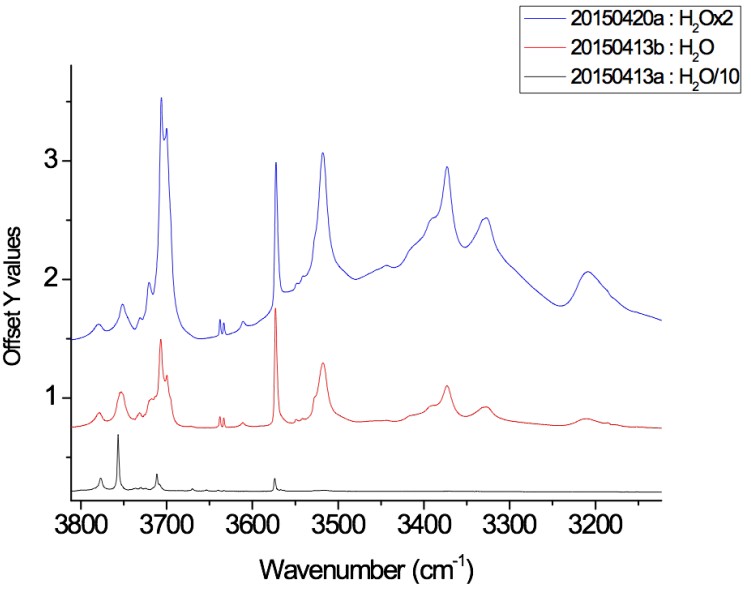

**Figure A3.** From ref. [18] Experimental FTIR spectra of $H_2O$:Ar depositions at 10 K (a) 1:500 (black, only water monomers and dimers are present); (b) 1:50 (red, larger water aggregates appear (trimers, tetramers, pentamers and hexamers); and (c) 1:25 (blue, a mixture of water aggregates and ASW).

## Appendix E. FTIR Bands of Corannulene in Argon Matrices at 10 K

**Table A6.** Observed bands positions ($cm^{-1}$) and relative intensities (vs: very strong, s: strong, ms: medium strong, m: medium, mw: medium weak, w: weak) of corannulene isolated in argon matrices at 10 K and comparison with results by Rouillé et al. [39] in an argon matrix at 12 K Note that spectra of Rouillé et al. [39] also contained small amounts of water.

| Vib. Modes | Wavenumbers ($cm^{-1}$, This Work) | from Ref. [39] |
|---|---|---|
| $\nu$(C-H) | 3064.5 m | |
| | 3056.1 m | |
| | 3050.5 m | |
| | 3043.0 m | |
| | 3037.4 m | |
| Combination modes | 1906 w | |
| | 1893 w | |
| $\nu$(C-C) and $\delta$(C-H) in-plane bending | 1441.0 m | 1442.7 |
| | | 1428.8 |
| | 1316.3 m | 1316.9 |
| | | 1239.3 |
| | 1138.8 mw | 1138.6 |
| | | 1025.2 |
| Out-of-plane bending $\gamma$(C-H)) | 837.1 vs | 837.2 |
| | 818.3 w | 817.5 |
| | 749.8 w | 748.7 |
| | 661.8 s | 661.5 |
| | 547.4 m | |
| | 446.8 w | |
| | 405.9 w | |

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
