# Peer review of "Water Clusters in Interaction with Corannulene in a Rare Gas Matrix: Structures, Stability and IR Spectra"

_2673-7256, doi:10.3390/photochem2020018_

Round 1

Reviewer 1 Report

The Manuscript by Heloïse Leboucher , Joëlle Mascetti , Christian Aupetit , Jennifer A. Noble, and Aude Simon, deals with the stability of corannulene -n H2O(n=1,2,3) systems emebedded in a Argon matrix. 

The work is well written and scientifically sound. Theoretical and computational methods have been competently applied and the conclusions well support the results. The information inferred from the results gives interesting insights into the stability of the system according to the boundary conditions. Moreover, the outcomes also help to (partially) infer the photoreactivity of the systems even if, as also stated by the authors themselves, the relevant excited electronic study is the missing piece.

In my opinion, the manuscript deserves to be published in its current form.

Author Response

see the .docx file

Reviewer 2 Report

This manuscript could be accepted for publication in Photochem. The novelty of presented research and it's importance for the scientific community of photochemistry are high. In this manuscript, authors report a combined theoretical and experimental study of corannulene-water interactions in low temperature matrices and of the matrix’s influence on the photoreactivity of corannulene with water. The introduction provide sufficient background. The research methodology is adequate and modern, it would be good idea to add in the manuscript some references on relevant works focused on spectroscopic studies of other chemical systems:

Biophysical Journal 2055, V. 88, P. 3829.; J. Mol. Struct. 2020, V. 1199. P. 126789.; J. Chem. Phys. 2016, V. 145, P. 154504.; J. Phys. Chem. A 2018, V. 122, P. 832.; J. Chem. Theory Comput. 2016, V. 12, P. 1905.; Inorg. Chim. Acta 2018, V. 482. P. 838.; J. Phys. Chem. C 2015, V. 119, P. 1768.; Phys. Chem. Chem. Phys. 2016, V. 18. P. 14104.; Phys. Chem. Chem. Phys., 2020, V. 22. P. 12785.; ChemistrySelect 2016, V. 3. P. 456.; Adv. Theory Simul. 2020, V. 3, P. 2000174.; Open Journal of Physical Chemistry 2012, V. 2, P. 228.

The results are clearly presented. The amount of data is large. The conclusions supported by the data. The manuscript good illustrated and interesting to read. English language and style are fine, and may be very minor polishing from native speaker is recommended. Finally, in the end of the manuscript there are some problems with formatting (references are mixtures with Appendices).
Overall, this nice manuscript could be accepted for publication after very minor revisions.

Author Response

see .docx file

Reviewer 3 Report

In the manuscript entitled "Water clusters in interaction with corannulene in a rare gas matrix: structures, stability and IR spectra" the authors present the results of combined theoretical and experimental study of corannulene-water interactions in low temperature matrices. The theoretical study was performed using a mixed density functional based tight binding/force field approach to describe the corannulene-water clusters trapped in an argon matrix, together with Born-Oppenheimer molecular dynamics to determine finite-temperature IR spectra. The results are discussed in the light of experimental matrix isolation FTIR spectroscopic data. In my opinion, all the calculations and simulations have been performed competently and in most cases the conclusions drawn are well supported by the experimental data. The substantial content of the manuscript as well as its general quality is above average and in the present form the manuscript seems to meet the general requirements justifying publication in PhotoChem, and should be accepted as it is.

Author Response

see docx file

Reviewer 4 Report

In this manuscript, the authors have performed matrix isolation experiments on water corannulene clusters and obtained their IR spectra.  The results were compared with DFTB based molecular dynamics to connect spectral changes with the geometrical structure of the clusters.  From the shifts in the OH stretching band of the water molecules in the water corannulene cluster, the authors conclude that no (C10H20)(H2O) complex is detected, while (C10H20)(H2O)3 with water trimer interacting with the concave side of the corannulene is identified.  They also concluded that they did not detect the interaction of corannulene with larger water clusters (C10H20)(H2O)n, n=4-6.  They use theoretical calculations to strengthen their claim, but I feel it is still weak.  I would like the authors to answer the following before accepting for publication.

As for the experimental conditions, how was the water concentration controlled?  In line 381, they mention they added water to the Ar, Coroannulene. Can they do a more thorough test on water concentration to try to produce the (C10H20)(H2O)?

The main conclusion is based on a comparison of the peak shifts of the OH stretching mode of H2O, and it seems a bit weak.  Can the authors also compare the difference spectra of the experiment given in Figure 9 with the difference spectra from the DFTB-MD?

Also, I am not so sure I understand why the authors claim larger water clusters (C10H20)(H2O)n, n=4-6 were not detected.  If they calculate the (C10H20)(H2O)4 IR spectra and compare that with (C10H20)(H2O)3 IR spectra, they can show distinct differences between them.  If this difference between 3 and 4 water clusters is not seen in the experimental spectra, I can accept that (C10H20)(H2O)4 is not there.  Now it is not conclusive, it can be that the spectra of (C10H20)(H2O)4 may be very similar to (C10H20)(H2O)3, and that is the reason you can not assign higher aggregation of water molecules.

Next, some discussion should be given for using only 7 trajectories.  The convergence of the spectra should be given.  For example, if the authors compare the spectra calculated by averaging 3 (or 5) trajectories and take the difference with those obtained by 7 trajectories, how does the difference spectra look?  Is the difference spectra converging to nearly flat spectra as we increase the number of trajectories?  In that case, we can say results are converged.

Minor issues are as follows:

Some verification of the accuracy of the DFTB with quantum chemistry calculations should be given.

Figures are randomly distributed in the manuscript.

How were the dipole moments calculated?

In line 136, the authors mention that the Ar is frozen in the simulation.  How was this done?  Did they use SHAKE algorithm, or was the motion left out of the propagation?  I feel this artificial freezing may induce additional shifts to the IR peaks.  Can the authors comment on this?

In lin 406, they mention the addition of UV photons did not lead to much change.  It will be great to correlate this result with the astrophysical studies mentioned in the intro.  Does this mean we should differentiate flat or curved PAH when we discuss the UV absorption and electron transfer reaction with H2O?

Author Response

Referee report 4: In this manuscript, the authors have performed matrix isolation experiments on water corannulene clusters and obtained their IR spectra.  The results were compared with DFTB based molecular dynamics to connect spectral changes with the geometrical structure of the clusters.  From the shifts in the OH stretching band of the water molecules in the water corannulene cluster, the authors conclude that no (C10H20)(H2O) complex is detected, while (C10H20)(H2O)3 with water trimer interacting with the concave side of the corannulene is identified.  They also concluded that they did not detect the interaction of corannulene with larger water clusters (C10H20)(H2O)n, n=4-6.  They use theoretical calculations to strengthen their claim, but I feel it is still weak.  I would like the authors to answer the following before accepting for publication.

Author reply. We thank the referee for his interest in our work and the issues he/she mentioned. We answer his/her comments below, point by point. All the modifications in the text of the new version of the manuscript are written in blue for the sake of clarity.

As for the experimental conditions, how was the water concentration controlled?  In line 381, they mention they added water to the Ar, Coroannulene. Can they do a more thorough test on water concentration to try to produce the (C10H20)(H2O)?

Author reply. As described in section 2.2 (experimental details), our H2O:Ar mixtures are prepared in a dosing line from partial pressures of water and argon in the range 1:10 to 1:1. The mixtures are then co-injected in the cryogenic chamber with the corannulene vapor and additional argon if we want to obtain more diluted water:argon mixtures (1:500 to 1:25). In fact, the control of the water concentration is given by what we observe in our FTIR spectra: presence of monomers only, or a mixture of monomers and dimers, etc… up to the formation of larger clusters and finally water ice (see Figure A3). It has been impossible to obtain a C20H10:Ar deposit without any trace of water (see lines 408 to 411 in the revised version of the manuscript), which was also the case with coronene (see Noble et al 2017, reference 17 in the new version of the manuscript), and is typical of hygroscopic molecules. In panel “a” of Figure 9 in the revised version of the manuscript the presence of water monomer and of a trace of dimer can be observed. As no difference has been observed on water lines, nor on corannulene lines in this spectrum, as well as on water monomer lines in C20H10:H2O:Ar deposits containing added water (H2O:Ar=1:50),  we conclude that no (C20H10)(H2O) is formed in our experimental conditions. This can be due to the effect of the argon environment, or to a better reactivity of corannulene with small aggregates (water dimer and trimer) compared to the water monomer (see discussion). We have modified the text lines 499-501 and 505-506 (in blue in the revised version) to clarify these points.

The main conclusion is based on a comparison of the peak shifts of the OH stretching mode of H2O, and it seems a bit weak.  Can the authors also compare the difference spectra of the experiment given in Figure 9 with the difference spectra from the DFTB-MD?

Author reply.  As some bare water remains in the C20H10:H2O:Ar mixture, using the experimental difference spectra between C20H10:H2O:Ar and H2O:Ar mixtures allows an easier identification of shifted or weak new bands and thus a safer assignment. Computing the difference spectra from  DFTB-MD is not an appropriate method in our opinion as the spectra of (H2O)n, (C20H10)(H2O)n and (C20H10) are computed separately. This is not an issue for calculations. Furthermore, direct comparison between experimental and theoretical difference spectra would be difficult as we do not use the calculated harmonic wavenumbers (which are always calculated too high in energy), but the calculated shifts to identify complexes formed in our matrices. This approach was previously benchmarked and found to be very relevant for many PAH-water clusters systems. To inform the reader about our previous works on these topics, and to provide detailed information on the theoretical approaches which were used in the paper, we added a whole paragraph in section 2.1 (see lines 112 to 135 and 139 to 142 in blue in the revised version of the manuscript).

Also, I am not so sure I understand why the authors claim larger water clusters (C10H20)(H2O)n, n=4-6 were not detected.  If they calculate the (C10H20)(H2O)4 IR spectra and compare that with (C10H20)(H2O)3 IR spectra, they can show distinct differences between them.  If this difference between 3 and 4 water clusters is not seen in the experimental spectra, I can accept that (C10H20)(H2O)4 is not there.  Now it is not conclusive, it can be that the spectra of (C10H20)(H2O)4 may be very similar to (C10H20)(H2O)3, and that is the reason you can not assign higher aggregation of water molecules.

Author reply. We have claimed that no (C10H20)(H2O)n (n>3) complexes have been formed because we do not see any experimental evidence for that on our difference spectra. The water tetramer and larger clusters spectra are quite different from those of water dimer and trimer and located lower in energy. So it is highly improbable that (C10H20)(H2O)4 has been formed. We have modified our text to explain more clearly this point (lines 506-510). Given the huge amount of work it would represent given the increasing number of isomers to consider, the computations for (C10H20)(H2O)4 have not been made.

Next, some discussion should be given for using only 7 trajectories.  The convergence of the spectra should be given.  For example, if the authors compare the spectra calculated by averaging 3 (or 5) trajectories and take the difference with those obtained by 7 trajectories, how does the difference spectra look?  Is the difference spectra converging to nearly flat spectra as we increase the number of trajectories?  In that case, we can say results are converged.

Author reply. We thank the reviewer for this comment on the convergence of the spectra. We have added a Figure (Figure A1 in the new version of the manuscript) that reports some convergence tests for the stretching bands of the water molecules and clusters for three systems investigated in the present paper, following the procedure suggested by the reviewer. We hope that it illustrates well that the convergence on positions is reached for 7*100 ps MDs whereas the convergence on intensity requires some additional times in some cases. The resolution of the spectra is 0.4 cm-1 therefore the noise is quite visible. We also added some text referring to this convergence test in the main manuscript (see lines 177-180).

Minor issues are as follows:

Some verification of the accuracy of the DFTB with quantum chemistry calculations should be given.

Author reply : as explained in the new text in blue in section 2.1 of the revised version of the manuscript, the methodology used in the present paper was developed in our group and carefully benchmarked on systems involving the same interactions and of similar size as the present one. As mentioned above, to inform the reader about our previous work on this topic, and to provide detailed information about the theoretical approaches used in the paper, we added a whole paragraph in section 2.1 (see lines 112 to 134 and 138 to 140). We also added some results comparing DFTB to DFT for specific corannulene-water complexes in the gas phase in the Zenodo database, to which a new doi was assigned (https://doi.org/10.5281/zenodo.6368179).

Figures are randomly distributed in the manuscript.

Author reply : we apologize for this inconvenience, due to issues with LaTeX formatting: it has been corrected.

How were the dipole moments calculated?

In line 136, the authors mention that the Ar is frozen in the simulation.  How was this done?  Did they use SHAKE algorithm, or was the motion left out of the propagation?  I feel this artificial freezing may induce additional shifts to the IR peaks.  Can the authors comment on this?

Author reply : we agree with the referee that the methodology was not sufficiently detailed in the previous version of the manuscript, therefore we added some text (in blue lines 112 to 136 and 139 to 142, and 187 to 189) in section 2.1 detailing some aspects of the methodology and explaining that the approach used in the present paper was developed by our group and carefully benchmarked on systems with interactions similar to the one studied in the present paper. In particular, CM3 charges are used for the corannulene:water impurity, and the dipole moment is computed with these charges (see lines 168 to 169). The Ar atoms are not charged and those which are fixed are very far from the impurity (5 layers of Ar atoms separate them) so we expect that  if the positions are frozen (not allowed to move at all), no additional shift in the IR spectra will be observed. Actually, this methodology was carefully benchmarked for water clusters inside the Ar matrix, as well as the size of the matrix needed for the IR spectra to be converged (see ref. 55 in the revised version of the manuscript). We added a new Figure 2 to help the reader figure out the size and shape of the studied theoretical systems.

The Ar atoms that are frozen correspond to the most external ones. As we have not taken part in the development of this part of the deMonNano code, we do not know exactly which algorithm was used to freeze the positions of the most external Ar atoms.  We obtained these results using the keyword MDCONSTRAINT and giving the positions of the atoms. The user guide mentions that using this keyword allows one to maintain the positions of the atoms fixed during the dynamics.

In lin 406, they mention the addition of UV photons did not lead to much change.  It will be great to correlate this result with the astrophysical studies mentioned in the intro.  Does this mean we should differentiate flat or curved PAH when we discuss the UV absorption and electron transfer reaction with H2O?

Author reply. In this work, we consider photoreactions in a condensed phase (argon matrix). We have shown that the complexes formed  are different from that observed in the gas phase by other authors (see Perez et al, 2017). As mentioned earlier, this can be due to the effect of the argon environment, which may favor interaction geometries. From an astrophysical point of view, we think that there is probably no difference between flat and curved PAHs when using high energy UV (ie Lyman alpha) that leads to PAH ionization, but that influence of mutual orientation and distance occurs when using low energy photons that induce charge transfer reactions, as it appears that the differentiation between flat and curved PAHs may be of importance in UV absorption and electron transfer reactions in PAH/water molecular systems. However, it is still premature to project the results obtained for coronene and corannulene directly to PAHs of astro-relevant sizes embedded in water ice.   This point has been added in the conclusion (lines 530-536).

Round 2

Reviewer 4 Report

The authors have answered all issues raised by the referee. I think the manuscript is well written and concise now. Therefore, I recommend it for publication.